# Short high-accuracy tritium data time series for assessing groundwater mean transit times in the vadose and saturated zones of the Luxembourg Sandstone aquifer

Laurent Gourdol[1], Michael K. Stewart[2], Uwe Morgenstern[2], Laurent Pfister[1,3]

[1]Luxembourg Institute of Science and Technology (LIST), Belvaux, Luxembourg
[2]GNS Science, Tritium & Water Dating Laboratory, Lower Hutt, New Zealand
[3]University of Luxembourg, Faculty of Science, Technology and Medicine, Belval, Luxembourg

*Correspondence to*: Laurent Gourdol (laurent.gourdol@list.lu)

**Abstract.** Among the manifold of environmental tracers at hand, tritium is the only one that can inform on groundwater age within the time scale of hundred years for the entire flow system, i.e. unsaturated and saturated. However, while in the Southern Hemisphere a single water sample is sufficient for tritium-based young groundwater dating, several tritium measurements spanning over multiple years are still needed in the Northern Hemisphere to disentangle the natural cosmogenic tritium input from that caused by the atmospheric thermonuclear weapon tests mainly carried out in the early 1960s. Although it is advised to focus tritium dating on sites where long chronicles of tritium data are available, we tested in this study the potential for short high-accuracy tritium data series (~4 years) to date groundwater from 35 springs draining the Luxembourg Sandstone aquifer (Central Western Europe). We determined groundwater mean transit times by using the lumped parameter model approach in a Monte Carlo uncertainty estimation framework to provide uncertainty ranges inherent to the low number of tritium data at hand and their related analytical errors. Our results show that unambiguous groundwater mean transit time assessments cannot be determined solely based on such recent short tritium time series, given that several ranges of mean transit times appeared theoretically possible. Nonetheless we succeeded in discriminating groundwater mean transit times in the vadose and saturated zones of the aquifer through a stepwise decision process guided with several supplementary data. The mean transit time required for water to cross the vadose zone was estimated to be between 0.5±0.5 and 8.1±1.2 years depending on the spring, while for water to flow through the saturated zone it varied from 5.7±2.4 to 18.9±4.6 years (median ± semi 5-95 percentile range). Our findings are consistent with both the tritium measurements of individual springs and the hydrogeological context of the study area. We specifically corroborated the dating results using the known hydrogeological properties of the Luxembourg Sandstone aquifer, the hydrochemistry of the studied springs as well as their discharge dynamics. When translated into water velocities (which average ~12 m/year and ~170 m/year for the vadose and the saturated zones, respectively), the tritium dating results mirrored the horizontal-vertical anisotropy of the aquifer's hydraulic properties caused by the bedded character of the Luxembourg Sandstone. In addition to improving our understanding of water transit times in the Luxembourg Sandstone aquifer, this study demonstrates how it is currently possible to use short tritium time series to date young groundwater bodies at new sites in Central Europe.

**Keywords.** Tritium, groundwater, mean transit time, Luxembourg Sandstone aquifer.

## 1. Introduction

Determining the timescales at which precipitation is percolating from a recharge area through an aquifer to where it discharges naturally (springs, streams) or is extracted from pumping wells is of key importance for a better understanding of groundwater resources and their sustainable management (Purtschert, 2008; Purtschert et al., 2008). Assessments of groundwater transit times are key to identify, model or forecast anthropogenic impacts of past and present land use practices on groundwater quality (e.g., MacDonald et al.,2003; Morgenstern and Daughney, 2012; Morgenstern et al., 2015). This is especially the case for young groundwaters, i.e., groundwater recharged during the last hundreds of years, including the time of change from low to high intensified land use. Young groundwaters constitute the most important component of the active hydrological cycle (Gleeson et al., 2015) and are more vulnerable and prone to surface-borne anthropogenic contamination than old groundwaters (Newman et al., 2010; Beyer et al., 2014; Jasechko, 2019).

The use of tracers – i.e., substances which can be detected in the water and allow following, or tracing, the flow of water – is particularly suitable to assess transit times of hydrological systems (Leibundgut et al., 2009; Leibundgut and Seibert, 2011). Tracers are divided into two general categories – environmental or artificial – depending on how they enter the system to be studied. While artificial tracers are purposefully injected over a limited temporal and spatial scale to infer transit times through preferential fast flow paths, environmental tracers – which, as their name implies, are present in the environment – make it possible to assess the overall and integrated response of hydrological systems (e.g., Einsiedl, 2005; Lauber and Goldscheider, 2014). The common feature of the environmental tracers used as dating tools is that they are time-dependent, whether because of variable input concentrations during recharge, chemical or physical accumulation processes in the subsurface and/or radioactive decay (Morgenstern et al., 2010).

Among the environmental tracers at hand, tritium (radioactive hydrogen isotope $^3$H) is particularly interesting for young groundwater dating (Newman et al., 2010; Beyer et al., 2014; Jasechko, 2019). Produced naturally in the atmosphere by cosmic-ray spallation processes (Lal and Peters, 1967), but also introduced artificially during the atmospheric nuclear weapons testing in the 1950's and 1960's (Morishima et al., 1985), tritium enters the water cycle as so-called "tritiated" water ($^3$H$^1$HO). Water from precipitation arriving at the ground surface thus contains tritiated water molecules in a given initial concentration, which then decreases by radioactive decay while in transit through the hydrosystem. Tritium decays with a half-life of 12.32±0.02 years (Lucas and Unterweger, 2000) and has the potential for dating groundwater with mean transit times up to about 100-200 years (Cook and Solomon, 1997; Morgenstern and Daughney, 2012; Beyer et al., 2014).

Alternative gaseous tracers covering a similar dating range to tritium exist (e.g., $^3$He, CFCs, SF$_6$, $^{85}$K) but their use has certain disadvantages (Morgenstern and Daughney, 2012; Cartwright et al., 2017; Gerber et al., 2018). Indeed, tritium, as part of the water molecule itself, allows inferring transit times including travel time through the unsaturated zone, whereas gas tracers dissolved in recharge water stay in equilibrium with the atmosphere in the unsaturated zone and derived dating results reflect only transit times in the saturated zone. Besides, tritium dating can be applied where an aquifer discharges naturally (springs,

streams), while the use of dissolved gas tracers at these places is challenging due to the risk of sample contamination with the surrounding air (Busenberg et al., 2006; Han et al., 2006).

The use of tritium within the hydrological sciences community for water dating purposes has evolved over time due to the varying origin and level of its atmospheric concentration. Even if the potential for natural-borne tritium for dating hydrological systems was suggested early on by Libby (1953), most of these pioneering applications capitalized on the considerable

precipitation tritium increases inherent to the mid-20[th]-century nuclear weapon tests (Jasechko, 2019). These elevated values allowed explicit water transit time analyses for several years past the mid-1960's bomb peak, the latter acting as a clear time marker. With the bomb tritium pulse decay observed after the 1970's, tritium dating of hydrological systems had to increasingly rely on time series with several measurements to deliver unambiguous results – eventually leading to a fading interest in tritium as a tracer of choice for water age dating (Stewart and Morgenstern, 2016; Stewart et al., 2021).

Present-day tritium activities in precipitation have recovered natural cosmogenic levels, i.e., similar to levels noted before the nuclear tests (Cauquoin et al., 2015; Terzer-Wassmuth et al., 2022). However, bomb tritium remains present in the hydrological cycle of the earth (Stewart et al., 2021). The interest in tritium water dating has recently gained new momentum (Morgenstern et al., 2010; Stewart et al., 2012). In the Southern Hemisphere, the remnant bomb tritium has indeed declined to levels lower than those found in modern rainfall (Tadros et al., 2014; Stewart et al., 2021), which allows now unambiguous dating of

hydrological systems from a single sample relying on the tritium's radioactive decay (Morgenstern et al., 2010; Cartwright et al, 2020).

However, in the Northern Hemisphere the remnant bomb tritium activities have not yet decayed to below those of current rainfall, because precipitation was much more strongly affected by nuclear weapons testing (i.e., the tritium bomb peak was two to three orders of magnitude higher than in the Southern Hemisphere; Stewart et al., 2021). Time series with several tritium

measurements are still needed here for an unequivocal dating of hydrological systems and it is therefore presently advised to carry out tritium water dating preferably at sites where past data are available and can be supplemented with analyses of recent samples, so that a longer time series can be obtained (Stewart and Morgenstern, 2016). Some recent studies have leveraged data from such sites (e.g., Stolp et al., 2010; Blavoux et al., 2013; Gallart et al., 2016; Jerbi et al., 2019), but the latter remain rare or even absent for most hydrological systems.

As tritium in the environment is at much lower levels at present than in the past, an important aspect that must be considered is the tritium measurement accuracy and its related impact on water dating results (Eastoe et al., 2012; Stewart et al., 2012; Stewart and Morgenstern, 2016). Recent methodological developments have fortunately improved the tritium detection sensitivity and measurement accuracy (Morgenstern and Taylor, 2009). The advantage of using these high-precision tritium analyses for obtaining more accurate water dating results have been documented for both the Southern Hemisphere and the

Northern Hemisphere (Stewart et al., 2010, 2012; Gallart et al., 2016).

Here, we rely on the tritium dating approach to inform on young groundwater transit times in the Northern Hemisphere for new study sites, i.e., which do not have historical tritium records, but for which we can nevertheless rely on recent high-accuracy tritium data. The aim of this study is to evaluate water transit times for several springs draining the Luxembourg

Sandstone aquifer. Widely used for the country's drinking water supply, but also impacted by anthropogenic pollutants (e.g., Bohn et al., 2011; Gourdol et al., 2013; Farlin et al., 2018, 2019), the Luxembourg Sandstone aquifer is the most valuable groundwater resource of Luxembourg and, as such, assessing and understanding of water transit times is of national interest. Following a lumped parameter model approach to assess mean transit times (Maloszewski and Zuber, 1982), we set out to corroborate and interpret the tritium dating results in combination with several supplementary data, i.e. the known hydrogeological properties of the aquifer, the hydrochemistry of the studied springs as well as their discharge dynamics. By testing the potential of short but high-accuracy recent tritium data time series, our work also provides new insights for dating of young groundwater bodies at new sites in Central Europe.

## 2. Study area

Located on the north-eastern boundary of the Paris sedimentary Basin, the Luxembourg Sandstone is a sandy sedimentary ensemble dating from the Lower Liassic and interposed in the normal marly-limestone Lias facies. Due to its intrinsic properties, structure, and positioning, it constitutes the main aquifer in Luxembourg hosting the most important groundwater resource of the country and its capital.

### 2.1. The Luxembourg Sandstone: geological and hydrogeological settings

The Luxembourg Sandstone (Fig. 1; li2 unit in Fig. 2), whose average total thickness is about 60-80 m, outcrops over an area of about 350 km², edged by a cuesta escarpment (e.g., Fig. 1ab), which, in Luxembourg alone, has a length of more than 200 km (Fig. 2; Bintz and Muller, 1965; Colbach, 2005; Kausch and Maquil, 2018). Outcropping to the northeast, beyond the German border, and expanding towards a broad plateau in the southern part of Luxembourg reaching out to Belgium, the Luxembourg Sandstone, gently dipping towards SW (<10°), dives below more recent geological strata in France. Structurally, as the Mesozoic sediments of Luxembourg, it is also characterized by mainly SW-NE striking horsts and grabens (main faulting direction, associated to large undulations) and affected by NW-SE conjugate faults (Berg, 1965; Colbach, 2005).

The Luxembourg Sandstone originated from offshore sandwaves which deposited on a flat shallow marine shelf. Structured into several sequences, these deposits have different subfacies, related to distinct evolution stages (Berners, 1983; Colbach et al., 2005; Schäfer and Colbach, 2021). It is worth noting that in Belgium the Luxembourg Sandstone formation is subdivided in different members separated by marl beds, while in Luxembourg marly interlayers exists only discontinuously and locally (Boulvain et al., 2000, 2001; Van den Bril and Swennen, 2009; Kausch and Maquil, 2018). Mineralogically, the bulk of the Luxembourg Sandstone is made of quartz grains bonded by a calcareous cement (Fig. 1c). Its average composition is about 65-70% quartz and 30-35% calcium carbonate, but due to a bimodal distribution of the carbonate content the Luxembourg Sandstone stands as a bedded alternation of two contrasted sandstone types (Fig. 1d), namely sandstone (poorly cemented yellowish sandstone with less than 10% carbonate) and calcareous sandstone (grey to whitish cement-rich sandy limestones with up to 60% carbonate), which are about equally represented in the formation (Colbach, 2005; Kausch and Maquil, 2018).

This changing carbonate content in the Luxembourg Sandstone formation has a primary synsedimentary, bioclastic origin, but was also influenced by secondary diagenetic redistribution processes (Colbach, 2005; for details see Berners, 1983; Molenaar, 1998; Van den Bril and Swennen, 2009; Schäfer and Colbach, 2021). The calcium carbonate content determines the variability of the porosity of the Luxembourg Sandstone, which varies from 5% in the sandy limestones up to 40% in the poorly cemented sandstones (Colbach, 2005; Kausch and Maquil, 2018; Claes et al., 2018; Samir, 2019).

Colbach (2005) reports a mean hydraulic conductivity for this formation of around $5 \times 10^{-5}$ m/s, but the hydraulic conductivity of the Luxembourg Sandstone is actually heterogeneous. Kausch and Maquil (2018) denote indeed that sandstone beds are highly permeable (hydraulic conductivity up to $10^{-2}$ m/s) while calcareous sandstone beds are of low permeability in comparison. The contrasting mechanical properties of sandstone and calcareous sandstone beds determine the crack density as well in the Luxembourg Sandstone. Owing to their high porosity, the sandstone beds are indeed much lighter and less resistant

to compression than the calcareous sandstones ones. Consequently, crack density is high in the calcareous sandstones and weak in the porous sandstones (Colbach, 2005; Kausch and Maquil, 2018; Fig. 1e).

    Independent of the small joints developed in the calcareous sandstone, large-scale fracture and joint patterns also affect the Luxembourg Sandstone (e.g., Fig. 1f). Related to the regional geological structure (i.e., SW-NE and NW-SE fault system), the Luxembourg Sandstone is indeed cut by a nearly vertical, sub-orthogonal network of primary joints with a spacing of one to

several meters, which define large blocks or slabs (Colbach, 2005; Kausch and Maquil, 2018). A secondary fracture system is also associated with the faults (Kausch and Maquil, 2018). These large-scale fractures and joints are moreover often opened by karstification process (i.e., solubilization of the carbonated cement by surface or subsurface weathering processes along fractures and stratification with a mechanical removal of the residual material; Kausch and Maquil 2018; Meus and Willems, 2021). Even if the degree of karstification of the Luxembourg Sandstone degree is low, it should be noted that Meus and

Willems (2021) suggest, on the basis of several field observations (i.e., dismantled conduits parallel to the valley axis, subhorizontal plurimetric pipes and cavities developed out of discharge fracturation, and residual crusts covering wall voids), the potential presence in the Luxembourg Sandstone of the remains of a deep karst system intersected and renewed by the surface and subsurface weathering processes.

    All of these properties confer on the Luxembourg Sandstone reservoir a multiple porosity and permeability structure which

controls the groundwater storage and circulation within it. Mainly supported through the pores of the sandstone beds, the storage of water also takes place in in the joint and fracture networks. Likewise, even if joints and fractures drain the aquifer, the groundwater is circulating through the pores of the sandstones as well (Von Hoyer, 1971; Kausch and Maquil, 2018). As typically observed in bedded aquifers in general (e.g., Maier et al., 2022) and in other sandstone bedded formations in particular (e.g., Hitchmough et al., 2007; Clavaud et al., 2008; Goupil et al., 2022), a key feature caused by the alternation of contrasting

sandstone types in the Luxembourg Sandstone is the anisotropy of its ability to allow groundwater to circulate: i.e., the permeability perpendicular to bedding planes (vertical permeability) is lower than horizontal permeability (parallel to bedding planes). Even if it is not quantified, this anisotropy was observed when tunnels were bored in the Luxembourg Sandstone (Fox et al., 2008). Statistical analysis of more than a hundred artificial tracer tests carried out in the Luxembourg Sandstone aquifer,

Meus and Willems (2021) have additionally demonstrated the key role of the NE-SW fracture network on the groundwater drainage system in the saturated zone. It is also worth noting that observed transit times to first arrivals and concentration peaks for positive tracer tests led Meus and Willems (2021) to assess average velocities of 52 and 16 m/h respectively (with highest observed velocity of 350 m/h). These figures are fundamental for the understanding of the Luxembourg Sandstone aquifer, but nevertheless remain informative only on its fissured fast flow component. Although this fast flow component is of key importance for the protection of the resource (especially for bacteriological contamination from the surface or following localized point source accidental pollution), it is assumed not to dominate water transit times in the aquifer at large scale (Farlin et al., 2013a).

Overall, the groundwater body within the aquifer in Luxembourg is unconfined to the NE, where the Luxembourg Sandstone outcrops in plateau position, while to the SW it becomes confined as the Luxembourg Sandstone dives below a powerful charge of newer and mostly impervious strata. Our study is focusing on the unconfined part of the Luxembourg Sandstone aquifer, which is perched on the impervious Elvange Marls (li1 unit in Fig. 2) and notched by several deep valleys. Mainly fed by direct infiltration from the top of the plateaus where the Luxembourg Sandstone outcrops, groundwater accumulates at the base of the sandstone reservoir (Fig. 1g). The significantly weathered state of the upper metres of the outcropping sandstone bedrock and the sandy nature of the overlying soils have limited the development of the river network and effective precipitation infiltration can be considered as the dominant process there (Cammeraat et al., 2018; Kausch and Maquil, 2018). Covered in places with the Strassen Marls and Limestones (li3 unit), the Luxembourg Sandstone aquifer is also fed there by groundwater leaking from this overlying formation. In valleys notching the Luxembourg Sandstone, where the sandstone's impermeable base lies above the valley floors, the aquifer's saturated zone outflows either diffusely or in concentrated form as springs. The location of the many contact springs delimiting the base of the sandstone (more than 500 in the country) is determined by the dip direction of the aquifer basis, which mainly controls the saturated zone groundwater flow together with the Luxembourg Sandstone fracture network (Colbach, 2005; Kausch and Maquil, 2018). It is worth noting that small, perched groundwater bodies may temporally form on top of the local marly interlayers intercalated in the Luxembourg Sandstone formation (Kausch and Maquil, 2018; Fig. 1g).

## 2.2. The springs of Luxembourg City

Most of the natural contact springs draining the Luxembourg Sandstone aquifer are tapped, representing the main groundwater resource for the country's drinking water supply (AGE, 2015). Fig. 3 sketches a typical geological cross-section illustrating the catchment facility of a spring draining the Luxembourg Sandstone unconfined aquifer and the water flow components within. Among the tapped springs, 62 are in the vicinity of Luxembourg City and provide about two-thirds of the drinking water supply of the capital's population. It is this strategic importance that has eventually stimulated research efforts on spring water transit time.

Based on the local geomorphology of the geological layer, the springs of Luxembourg City can be grouped into eight main districts (Fig. 2): Birelergrund (B), Glasburen (G), Dommeldange (D), Kopstal rive droite (KRD), Kopstal rive gauche (KRG),

Millebach (M), Pulvermuhle (P) and Siwebouren (S). Within a national effort for improving the protection of groundwater bodies used for drinking water supply, the recharge areas of each of these groups have been redefined recently (RGD, 2018, 2019, 2021a, 2021b, 2022a, 2022b). Their delineation was based on water balance calculations together with constraints defined by the geological, geomorphological, and hydrogeological contexts (AGE, 2010). The mean outflowing discharge used to estimate the extension of the recharge areas (RGD, 2018, 2019, 2021a, 2021b, 2022a, 2022b) as well as the mean elevation and the surface of each recharge area are provided in Table 1. One can notice that the Luxembourg Sandstone is mostly outcropping in the study area. It is overlaid with the Strassen Marls and Limestones formation (Fig. 2) solely in the recharge areas of KRD and S groups, but in a very small part (~5%).

Hydrogeological drillings in the Luxembourg Sandstone are too sparse and poorly distributed for establishing an accurate water table map which would have enabled either to directly draw groundwater flow paths in the saturated zone or to estimate directly saturated and unsaturated zone thicknesses. However, an estimation of the average groundwater flow lengths was made for each group of springs by considering the distances between the outer limit of the recharge area and the emergence line defined by the springs. The Luxembourg Sandstone thickness was also computed by subtracting the terrain's surface topography from that of the sandstone's impermeable base (interpolated from the elevation of the known geological contact lines). These two attributes are also listed in Table 1.

Thanks to an existing borehole drilled in the KRD area, it is finally worth noting that a marly interlayer exists there in the Luxembourg Sandstone. In this location, thicknesses of 10 and 45 m were also reported for the saturated and vadose zones, respectively (Farlin et al., 2013a).

## 3. Material and methods

### 3.1. Tritium water dating

#### 3.1.1. Transit time distribution and mean transit time estimation

Groundwater discharging at a spring is a mixture of water from short and long flow lines, and therefore the water in a sample does not have a single age corresponding to a unique transit time within the aquifer (i.e., related to the time elapsed since the water molecules entered the system at the recharge area, where the age is zero), but has a transit time distribution (TTD; Maloszewski and Zuber, 1982; Stewart et al., 2017). A mean transit time may however be estimated using the Lumped Parameter Model (LPM) approach. Introduced in hydrogeology by Maloszewski and Zuber (1982), LPMs are the most widely used model-based environmental tracer interpretation approach in groundwater studies (Turnadge and Smerdon, 2014). This method relates the concentration of a tracer (in our case tritium) at the outlet of a groundwater system ($C_{out}$) to its input concentration ($C_{in}$) using the well-known convolution integral (Maloszewski and Zuber, 1982):

$$C_{out}(t) = \int_0^\infty C_{in}(t - \tau) exp[-\lambda(\tau)] \, g(\tau) d\tau \qquad (1)$$

where $t$ is calendar time and the integration is carried out over the transit times $\tau$, $\lambda$ is the decay rate of the tracer ($\lambda = 0$ for stable tracers; $\lambda = \ln2/T\frac{1}{2}$ for radioactive tracers where $T\frac{1}{2}$ is the half-life; tritium $T\frac{1}{2} = 12.32$ years), and $g(\tau)$ is a transfer function defining the TTD.

The TTD should be described by a conceptual flow or mixing model reflecting the average (steady-state) conditions in the groundwater system (Maloszewski and Zuber, 1982; Leray et al., 2016). Within this work we choose the exponential-piston flow model (EPM; Maloszewski and Zuber, 1982) which mimics a system with two segments of flow in series: one of exponential flow and the other of piston flow. EPM can be used especially when the time lag required for water to flow through the unsaturated zone of an unconfined aquifer cannot be neglected (Jurgens et al., 2012). This is the case for the Luxembourg Sandstone areas that feed the springs of Luxembourg City (i.e., due to the unsaturated zones being expected to be thick for some areas while they are thin for others, as well as to the low vertical permeability relative to the horizontal one). In this context, the piston flow component simulates the water flowing through the unsaturated zone, while the exponential model represents the mixing at the outlet of groundwater flowlines coming from the saturated zone. The EPM response function is given by:

$$g(\tau) = 0 \quad for \quad \tau < \tau_m(1 - F) \tag{2a}$$

$$g(\tau) = \left(\frac{1}{F\tau_m}\right)\exp\left(-\frac{\tau}{F\tau_m} + \frac{1}{F} - 1\right) \quad for \quad \tau \geq \tau_m(1 - F) \tag{2b}$$

where $\tau_m$ is the mean transit time, and $F$ the ratio of the exponential volume to the total volume of the groundwater system. While the model is fully exponential when $F = 1$, it completely turns to piston flow when $F = 0$. It is also worth noting that $\tau_m$ can be split into two components, i.e., $\tau_{pf}$, the time lag required for water to flow through the piston flow section, and $\tau_{exp}$, the mean transit time for water to flow through the exponential volume. $\tau_{pf}$ and $\tau_{exp}$ are obtained by multiplying $\tau_m$ by $(1 - F)$ and $F$, respectively.

### 3.1.2. Spring water sampling and analytical techniques

Spring water samples were collected from 35 of the 62 springs of Luxembourg City (Fig. 2). The spring's selection was balanced to represent the main hydrogeological districts used for the water supply of the country's capital and to reflect a rather wide variability in terms of discharge dynamics and hydrochemical signatures (see 3.2 and 3.3). Note that some "springs", namely S01, S02, S03 and P01, are old natural emergences, but which have since then been excavated and are now being exploited through pumping.

To make the assumption of steady-state as acceptable as possible, samples were always taken outside of recharge and rainy periods in order to avoid any potential mixing with young freshwater coming from precipitation, e.g., through preferential pathways. All springs were sampled three times: in June 2011, August 2015 and 2017 for springs K21 and K21A and in

September 2013, August 2015 and 2017 for all the others. Samples were collected without filtration or preservation in a 1 L plastic bottle filled to the top.

Tritium activities were measured on water samples that had been vacuum distilled and electrolytically enriched 95-fold using liquid scintillation spectrometry (Morgenstern and Taylor, 2009). For all samples, the reproducibility of the enrichment was 1% and the limit of detection was 0.02 tritium units (TU; 1 TU represents a $^{3}H/^{1}H$ ratio of $10^{-18}$). The measurement uncertainty

($1\sigma$), mainly due to the uncertainty of the enrichment and statistical errors of the ultra-low-level decay counting, ranges between 0.07 and 0.12 TU (1.2-1.9%), depending on the tritium concentration of the samples. These are the measurements referred to as high-precision measurements in our study.

### 3.1.3. Input time series

Monthly values of tritium in precipitation (including uncertainties) come from the GNIP (Global Network of Isotopes in

Precipitation) measurement network and were obtained from the WISER database (Water Isotope System for Data Analysis, Visualization, and Electronic Retrieval; IAEA (International Atomic Energy Agency) and WMO (World Meteorological Organization), 2019; https://nucleus.iaea.org/wiser). The data used from 1978 to present are from the Trier station in Germany (Stumpp et al., 2014; Schmidt et al., 2019), located only 40 km away from Luxembourg City. The history of tritium in precipitation prior to 1978 has been reconstructed through a log-log linear regression ($r^2 = 0.65$; p-value $< 10^{-15}$) by combining

records from Ottawa, Canada, from 1953 with those from Vienna, Austria, from 1961 (Jurgens et al., 2012; Stewart et al., 2017). Before 1953, the pre-bomb background concentration of tritium was estimated to be 8 TU (Jurgens et al., 2012).

To account for losses due to evapotranspiration (McGuire and McDonnell, 2006), the input of tritium to the ground as recharge was calculated through a weighting procedure like the ones reported by Stewart et al. (2007), Ozyurt et al. (2014) or Duvert et al. (2016). Here, we relied on the Thornthwaite's method (Thornthwaite, 1948) which is a water balance of the rootzone

performing monthly book-keeping of precipitation, evapotranspiration, and soil moisture. This approach is widely used in hydrogeology for estimating aquifer recharge (e.g., Lanini et al., 2016; Mammoliti et al., 2021). Basically, infiltration below the root zone (i.e., precipitation in excess of that required to supply actual evapotranspiration and the soil water store: effective precipitation) occurs only when the soil maximum water-holding capacity (i.e., the field capacity) is exceeded. The Thornthwaite's method also comes with an empirical potential evapotranspiration formula using only monthly mean air

temperature as input. Note that Pfister et al. (2017) compared Thornthwaite's monthly potential evapotranspiration data with that obtained via the Penman–Monteith (Allen et al., 1998) formula in a region encompassing our study area and found a difference of less than ±5%.

For our study, we set the soil maximum water-holding capacity to 100 mm which corresponds to an overall mean value of soil profiles covering the Luxembourg Sandstone (Hissler et al., 2015; Hissler and Gourdol, 2015). Given the relatively restricted

spatial extension of our study area, the monthly water balance calculation to obtain the effective precipitation was done with the precipitation and temperature data recorded at a single representative location, i.e. the Findel airport WMO weather station (WMO ID 06590; https://www.meteolux.lu). In operation since 1947, this meteorological station is less than 10 km away from

each spring of Luxembourg City (Fig. 2) and its elevation (368 m) is close to those of the spring recharge areas (Table 1). One can also notice that our choice to consider the effective precipitation spatially homogeneous at this work scale is also supported

by Pfister et al. (2017) who described the spatial variability of the precipitation as rather small over a region otherwise even wider than that containing our study area.

Instead of calculating single annual tritium recharge input values, a one-year sliding window was then used to obtain monthly values as follows:

$$C_{in\ i} = \frac{\sum_{j=i-11}^{i} C_j Peff_j}{\sum_{j=i-11}^{i} Peff_j} \qquad (3)$$

where $C_{in\ i}$ is the monthly tritium recharge for the $i$ th month, and $C_j$ and $Peff_j$ are the monthly values of tritium in precipitation and monthly effective precipitation amounts for the $j$ th month, respectively. A Tukey filter (Tukey, 1968) with coefficient 0.4 was applied to the moving window to avoid edge effects (Duvert et al., 2016). To obtain the respective monthly estimates,

note that the same weighting procedure was applied to both tritium in precipitation and their uncertainties.

### 3.1.4. Modelling strategy

A modelling strategy able to account for the propagation of analytical errors is advocated for transit time determinations using LPMs, especially when only a limited number of samples is available (Gallart et al., 2016), as is the case for the springs of Luxembourg City. To this end, instead of using a best-fit calibration approach, EPM parameters were investigated within this

work using a Monte Carlo sampling routine in a generalised likelihood uncertainty estimation framework (GLUE; Beven and Binley, 1992, 2014).

The approach used follows nearly the same modelling method proposed and applied in Gallart et al. (2016). The EPM parameter space ($\tau_m$, $F$) was randomly explored until enough behavioural solutions were found (5000 within this work). The goodness of fit of the simulations was evaluated using the Nash–Sutcliffe efficiency coefficient (NSE; Nash and Sutcliffe,

1970). To propagate tritium analytical errors, input (recharge) and output (spring) normally replicated time series were generated beforehand using the means and standard errors of every tritium measurement. These replicated values served for the assessment of the goodness of fit of every simulation.

### 3.1.5. Groundwater volumes

The volume of groundwater $V$ stored within an aquifer (also known as the turnover volume, the volume of mobile water or the

effective aquifer's volume) can be related to the mean transit time $\tau_m$ of the water leaving it and its average outflowing discharge $Q$ (Maloszewski and Zuber, 1982; McGuire and McDonnell, 2006; Morgenstern et al., 2010; Zhou et al., 2022):

$$V = Q \cdot \tau_m \qquad (4)$$

Calculating such $\tau_m$ - derived volumes will be used here to ensure that EPM solutions are to some extent physically valid regarding basic geometries and properties of the studied system.

### 3.2. Response time of spring discharge to precipitation input

### 3.2.1. Cross-correlation analysis applied to groundwater time series

Introduced in karst hydrology by Mangin (1981a, 1981b, 1984), the use of correlation analysis applied to groundwater time
series has since then widely expanded to understand the functioning of groundwater systems with varying characteristics. The cross-correlation analysis, which consists in studying the correlation between two time series (Box and Jenkins, 1970), was used in particular to assess the response of aquifer systems at springs or wells to precipitation input which is responsible for their recharge (e.g. Padilla and Pulido-Bosch, 1995; Angelini, 1997; Larocque et al, 1998; Lee and Lee, 2000; Panagopoulos and Lambrakis, 2006). Calculating the values of the cross-correlation coefficient results in a graph called a cross-correlogram
that represents the correlation as a function of the time lag between the two series. A usual precipitation-groundwater time series correlogram reaches a peak of correlation after a certain time lag, the latter characterizing the response time of groundwater to the precipitation input. This response time can then be interpreted in terms of flow mechanisms. For instance, response times of unconfined aquifers derived from cross-correlation analysis have often been related to unsaturated zone characteristics (e.g., Lee and Lee, 2000; Lee et al., 2006; Fiorillo and Doglioni, 2010; Bloomfield and Marchant,2013;
Bloomfield et al.,2015; Al-Jaf et al., 2021).

Here, we did not follow the standard cross-correlation method to define the response time of groundwater to precipitation input, but rather used the methodology originally proposed by Fiorillo and Doglioni (2010) and subsequently taken up by Bloomfield and Marchant (2013), Bloomfield et al. (2015) or Van Loon et al. (2017). Starting from the premise that groundwater response at a given time does not result from a single precipitation pulse at a previous moment, but that it depends
on a previous period of precipitation, Fiorillo and Doglioni (2010) propose an alternate way to the usual precipitation-groundwater time series cross-correlation analysis. Instead of using one single input time series, they suggest exploring multiple precipitation time series data constructed over varying cumulative time windows as input and retain the one with the maximum correlation peak. The correlation peak time lag of the considered cross-correlogram, added in a second step to the extent of the cumulative precipitation time window, thus allows to estimate a minimum and a maximum response time of the
groundwater system, respectively. The average of these two values is defined in this study as the mean response time MRT.

### 3.2.2. Spring discharge time series

Discharge of the springs of Luxembourg City has been measured from the mid-1990s, typically at weekly to quarterly time steps, with the aim to grasp both seasonal and multi-year temporal dynamics. These raw data were aggregated to monthly

mean values. If no observations were available for a given month, then a linear interpolation was used to estimate the missing monthly discharge values. Only gaps of maximum 6-month length were filled.

Monthly discharge time series were thus created for each investigated spring, except for P01 whose water collection facility is unsuitable for discharge measurements. Note that the springs S01, S02 and S03 were not processed individually, but jointly, because only records of their cumulative discharge are available. The length of the monthly discharge chronicles is 14 years or more, except for the M01 spring for which the time series is 7 years long. In 81% of the springs, observations span over 25 years. Finally, it is worth noting that on average missing, interpolated and measure-derived monthly discharge data relate to 4, 25 and 71% of the time series, respectively.

### 3.2.3. Input time series

In line with Jemcov and Petric (2009), who have documented the value of using effective infiltration in cross-correlation analysis to assess the functioning of several springs and their aquifers, we did not select precipitation as input signal, but used instead effective precipitation. The proportion of precipitation eventually percolating to the groundwater is indeed a function of several processes linked to climatic conditions, vegetation, and soil characteristics, and prevailing in the superficial section of the system. By using precipitation as the entry signal, these processes, which usually only affect the first few tenths of centimetres of the subsoil, are not separated from the processes deeper inside the aquifer. Using effective precipitation instead can overcome this oversimplification and potentially avoid misinterpretation if one is particularly interested in the internal structure of the aquifer.

As with the weighting of the tritium input time series, the calculation of the effective precipitation relied on precipitation and temperature data recorded at the Findel airport weather station and a soil water balance following the Thornthwaite's method (Thornthwaite, 1948; see 3.1.3 for details). Ultimately, we built a data set of 60 input time series for an effective precipitation accumulation period varying from 1 to 60 months.

### 3.2.4. Assessment of mean response times and uncertainty ranges

The cross-correlation analysis applied in groundwater studies typically considers multi-year time series data to characterize input-output relationships over the whole data period. An alternative moving window cross-correlation approach was recently proposed by Delbart et al. (2014) to assess temporal variability in response times. This increasingly used method (e.g., Cai and Ofterdinger, 2016; Jeong et al., 2017; Le Duy et al., 2021) was initially set to reveal the seasonal variability of the response time of an aquifer to rainfall by using a relatively short sized moving window (Delbart et al., 2014). Here, we applied the cross-correlation in the same moving window framework, but used a larger window (i.e., 5-year long window and 3-month moving increment) to attribute to springs not only a unique MRT to precipitation input but also an uncertainty range. To do so, the cross-correlograms between the multiple effective precipitation inputs and the discharge output for a given spring were first calculated for each moving window and the related MRT was defined as per section 3.2.1. Next, the results of individual

moving windows were collected as an ensemble of possible solutions and summarised using standard percentile values (i.e., $5^{th}$, $25^{th}$, median, $75^{th}$ and $95^{th}$).

## 3.3. Spring hydrochemistry

The chemistry of groundwater varies because of a wide range of natural and anthropogenic processes. Some hydrochemical variations are due to time-dependent water-rock interactions inside the aquifer and can be related to groundwater transit time

(Purtschert, 2008). For instance, major ion concentrations are sometimes found to increase with increasing groundwater transit time (e.g., Edmunds and Smedley, 2000; Katz et al., 2004; Zuber et al., 2005; Edmunds and Shand, 2009), although these relationships may be of complex nature (Morgenstern et al., 2010).

Since late 2004, the springs of Luxembourg City are regularly sampled for hydrochemical analyses (Bohn et al, 2011; Gourdol et al., 2013;). Among the hydrochemical parameters available, major ion concentrations ($Ca^{2+}$, $Na^{+}$, $Mg^{2+}$, $K^{+}$, $HCO_3^{-}$, $SO_4^{2-}$,

$Cl^{-}$), as well as dissolved silica ($SiO_2$), electrical conductivity ($EC_{25°C}$), dissolved oxygen ($O_2$), and pH have previously been related to water-rock interaction processes occurring inside the Luxembourg Sandstone aquifer (Von Hoyer, 1971; Gourdol et al., 2013). In this study, we thus relied on these parameters to possibly corroborate the tritium dating results and assessed as well their potential for serving as geochemical proxies of groundwater transit times that might be used at sites where no tritium data is available.

Sampling methods and analytical details related to the acquisition of these data are described in Gourdol et al. (2013). As the spatial variability of the hydrochemistry of the springs of Luxembourg City is stable over time, both year-to-year and seasonally (Gourdol et al., 2013), the available data were aggregated into a single average value for each parameter of all of the 35 springs considered in this study.

## 4. Results

### 4.1. Tritium water dating

### 4.1.1. Tritium in precipitation, recharge, and spring waters

Tritium analytical results for the investigated springs of Luxembourg City are reported in Table 2 and displayed in Fig. 4, together with the monthly and 1-year moving average tritium concentrations observed in precipitation at Trier, as well as the derived recharge input function.

Due to the mid-20th-century nuclear weapon tests, tritium in precipitation and recharge peaked at over 1000 TU in 1963 and then decreased progressively back to pre-bomb background level in recent years (Fig. 4a). We can notice that because precipitation is mostly effective in winter and rainfall tritium content is overall lower during this season, the tritium concentration in the estimated recharge input function is most of the time significantly below that determined for yearly mean rainfall.

Tritium concentrations in spring waters average 5.42 TU and range from 4.31 to 7.13 TU during the observation period. They display a clear general downward trend (average decrease of about 0.25 TU per year), while measured precipitation and derived recharge tritium time series have been relatively stable for a decade disregarding seasonal variation (2010-2019 mean value of 8.39 and 6.75 TU, respectively; Fig. 4b).

As shown in Fig. 4c, springs belonging to the same recharge area are generally characterized by similar tritium contents. For instance, the highest tritium concentrations are typically associated with group B springs and the lowest ones to springs from groups KRD and KRG. Note that tritium contents in group G are more heterogeneous from one spring to another.

### 4.1.2. Mean transit time simulation of spring waters

### 4.1.2.1. EPM parameter space exploration

The assessment of spring mean transit times using the tritium recharge input function through the EPM model was done in two stages as illustrated in Fig. 5 for one of the investigated springs.

Following previous tritium dating studies (e.g., Morgenstern et al., 2010; Stewart et al., 2010, 2012; Gallart, et al., 2016; Gusyev et al., 2016; Stewart and Morgenstern, 2016), we explored first the widest parameter space to which tritium allows access, i.e. with $\tau_m$ ranging from 0 to 200 years and $F$ between 0 and 1 (Fig. 5ab). An NSE threshold value of 0 was retained at this stage to define behavioural solutions. For every spring, one large subpopulation of solutions with $\tau_m$ values greater than 35 years was easily discriminable. This subpopulation, whose mean $\tau_m$ value varies between 64 and 98 years depending on the springs, can be confidently rejected as being unreasonably old given the characteristics of the aquifer (aquifer perched on an impermeable underlying layer and split by several valleys, isolated from potential connection with deeper or more extended aquifer layers), the discharge of the springs of Luxembourg City and the extension of their recharge areas. These $\tau_m$ values would indeed have required groundwater levels to be too close to or even above the land surface (according to Eq. (4); see details in 4.1.3).

The parameter space was thus narrowed in a second step with $\tau_m$ ranging from 0 to 35 years (bounds of 0 and 1 for $F$ were kept; Fig. 5cd). A more stringent NSE threshold value of 0.5 was also retained for this screening. Modelling results of this second screening are provided for each spring in the Supplement (Fig. S1-S35). Owing to the short record of tritium measurements for every spring and despite the reduced parameter space combined with the more severe NSE threshold, the 5000 resulting behavioural solutions were not distributed around an unambiguous peak but spread into several subpopulations, most of the time overlapping, and thus leading to an ensemble with a complex multimodal character. To obtain a more coherent ensemble of solutions, we decided to keep only the most likely third of the behavioural solutions by relying on a 2D kernel NSE weighted density thresholding filter (Fig. 5c, Fig. S1a-S35a). In practice, using this filter allowed the multimodal distribution to retain the (almost) unimodal set of solutions with respect to both EPM parameters with the highest probability, and it provided acceptable results for all springs. For most springs, the remaining behavioural solutions were circumscribed in a single area of the parameter space. For cases with more than one population, we ultimately restricted the most likely

behavioural solutions to the dominant one, i.e., both the best and largest subpopulation. It is worth noting that the decision to keep only a third of the samples was taken, after some trials, as a good compromise on the one hand to reject the least possible solutions and to limit cases requiring rejection of a secondary weaker subpopulation on the other hand. One can finally notice that in all cases an ensemble of more than 1000 behavioural solutions was retained.

### 4.1.2.2. Dating results

The retained ensembles of the most likely model solutions are summarized for every spring in Table S1 of the Supplement using standard percentile thresholds (i.e., 5th, 25th, median, 75th and 95th) and in Fig. 6 as violin and box plots. To facilitate the assessment of inter-spring variability, two additional scatter plots are presented in Fig. 7, i.e., the first relating $F$ and $\tau_m$ values and the second one $\tau_{pf}$ and $\tau_{exp}$ values.

Ranging from 0.68 to 0.93, NSE median values of the ensembles are on average 0.86 and above 0.8 for 31 of the 35 studied springs. This indicates an overall good agreement between modelled and measured tritium concentrations in spring waters. Predicted EPM parameters of springs are characterized by median values ranging from 0.46 to 0.97 for $F$ and between 7.4 and 24 years for $\tau_m$. Considering the difference between their 95th and 5th percentiles bounds as a metric, the ensembles of most likely model solutions display a degree of uncertainty varying between 0.07 and 0.53 for $F$ (mean 5-95th percentiles range of 0.29) and from 3.5 to 15.8 years for $\tau_m$ (mean 5-95th percentiles range of 8.6 years).

Considering the quite large overlap between ensembles in the parameter space, the predicted EPM parameters show similar values for springs belonging to the same recharge area on the one hand, while they also highlight significant differences between spring groups on the other hand (Fig. 6-7).

Springs belonging to groups S, M and D are characterized with an exponential to total flow ratio $F$ close to 0.7 (median values between 0.64 and 0.76) and relatively low $\tau_m$ values (median values ranging from 7.6 to 10.9 years). Group B and P springs stand out with higher median values, spanning from 0.77 to 0.97 for $F$ and between 14.1 and 19.3 years for $\tau_m$. Results obtained for group G are more heterogeneous, especially with respect to the mean transit time – the group indeed includes the spring with the lowest $\tau_m$ median value (C05, 7.4 years) and the one with the highest (C01, 24 years) – but also regarding the $F$ parameter (spring $F$ median values range from 0.67 to 0.83). Finally, characterized by $\tau_m$ median values ranging from 10.7 and 15.1 years, springs of the KRD and KRG groups are distinguishable from those located in other areas with particularly low $F$ values (median values between 0.46 and 0.67), indicating a greater importance of the EPM piston component in this region. Note that it is eventually for the KRD and KRG groups that the highest $\tau_{pf}$ median values are observed (up to 8.1 years).

Note that $\tau_m$ median values characterizing the retained ensembles of EPM solutions correlated positively and significantly with spring mean tritium contents (r = 0.62; p-value < 10$^{-4}$). The correlation between spring mean tritium contents and $\tau_{exp}$ median values is even stronger and more significant (r = 0.93; p-value < 10$^{-14}$).

### 4.1.3. Groundwater volumes

The volumes of water $V$ stored in the Luxembourg Sandstone aquifer that contribute to the discharge of spring groups have been estimated from Eq. (4) using the mean discharge values characterizing each recharge areas (Table 1) and the average of $\tau_m$ median values (Table S1) of springs belonging to each group.

Considering the average of $F$ median values (Table S1) of each group of springs, the calculated $V$ volumes were subsequently split in $V_{pf}$, the part of the water volume associated with the EPM piston component ($V_{pf} = (1 - F) * V$), and $V_{exp}$, the water volume stored in the EPM exponential section ($V_{exp} = F * V$). For each recharge area $S$ (Table 1) and assuming an overall porosity of 25% for the Luxembourg Sandstone (which was set regarding the available porosity figures values available from the literature, cf. 2.1), $V_{pf}$ and $V_{exp}$ have been finally converted into mean equivalent sandstone layers required for the water storage in the saturated zone (assumed to match the exponential part of the EPM model; $H_{exp} = V_{exp}/(S * 0.25)$) and the vadose zone (assumed to correspond to the EPM piston segment; $H_{pf} = V_{pf}/(S * 0.25)$).

The obtained results are displayed in Fig. 8 as a bar plot in which $H_{exp}$ and $H_{pf}$ calculated values are overlaying the mean thickness of the Luxembourg Sandstone estimated for each recharge areas (Table 1). Overall, Fig. 8 indicates that the retained EPM solutions are physically valid whatever the recharge area. In addition, the values obtained for the KRD area are consistent with the thicknesses of the vadose and saturated zones reported by Farlin et al. (2013a) in this region. It should nevertheless be noticed that results obtained for group B appear less optimal compared to those of the other spring groups. Indeed, according to the figures, the void filling rate by water in the vadose zone would reach 65% in this area, while elsewhere it ranges more realistically between 3 and 17% (mean value of 10%).

### 4.2. Response time of spring discharge to effective precipitation input

Results of the spring discharge – effective precipitation cross-correlation analysis are reported for each spring in Table S2 of the Supplement through standard percentile values and displayed in Fig. 9 as violin and boxplots.

As indicated by correlation peak values, spring discharge correlates generally well with accumulated effective precipitation regardless of the considered spring (medians of maximum cross-correlation peak values range from 0.7 to 0.9 and average 0.84; percentiles 5th and 95th are for most of the ensembles of solutions greater than 0.6 and 0.9, respectively). Whereas the maximum correlations are associated with widely varying extents of the cumulative effective precipitation window (median values of ensembles ranging from 4 to 37 months), observed correlation peak time lags are generally null or less than 4 months. The resulting MRT ensembles are characterized by median values spanning from 2 to 20 months and mean interquartile and 5-95th percentiles ranging from 8 to 18 months, respectively. The lowest MRT values are typically associated with group B springs and the highest ones to groups' KRD and KRG springs.

Finally, Fig. 10 compares the different spring discharge responses to the effective precipitation inputs according to their median MRT value. While springs with a median MRT lower than or equal to 10 months show a discharge dynamic characterized with a strong seasonal pattern, the ones with median MRT comprised between 10 and 15 months are characterized

predominantly by interannual dynamics. The overall discharge dynamic of the springs with a median MRT equal to or greater than 15 months is even more damped and delayed compared to the monthly effective precipitation input function.

### 4.3. Spring hydrochemistry

Spring mean physicochemical characteristics are provided in Table S3 of the Supplement and presented in Fig. 11 as a heat map where dark red and dark blue tones denote the highest and lowest values for each parameter, respectively.

Springs belonging to the same recharge area generally have very similar characteristics. For instance, group B springs are typically the most mineralized, while those of group KRD count among the less enriched (Fig. 11). The observed mineralization of spring waters and their hydrochemical facies are in line with previous observations (Von Hoyer, 1971; Gourdol et al., 2013) and reflect the dissolution of the calcareous cement of the Luxembourg Sandstone and its composition, the weathering of the grains of quartz of the sandstone and the oxidation of pyrite disseminated in the rock matrix.

Note that $SiO_2$, $Ca^{2+}$, $HCO_3^-$, $K^+$, total dissolved solids (TDS; sum of major ions and dissolved silica contents) concentrations and $EC_{25°C}$ measurements are strongly correlated with each other (minimum correlation r of 0.77). Due to second-order natural or anthropogenic factors (e.g., road salt inputs), $Na^+/Cl^-$ and $Mg^{2+}/SO_4^{2-}$ ratios are partly disconnected from the global enrichment of spring waters. It is finally worth considering that the observed depletion of $O_2$ and pH with increasing mineralization is also linked to the water-rock interaction (i.e., due to the pyrite oxidation).

### 525 4.4. Comparison of tritium derived transit times with other spring characteristics

We tested the correlation of tritium derived transit times with other spring characteristics by applying a simple linear model. Although this analysis would not identify direct causalities and does not account for possible multi-collinearities, it should nevertheless help to shed new light on potential controls of mean transit times.

The parameters involved here are $\tau_m$, $\tau_{exp}$ and $\tau_{pf}$ median values of the ensembles of EPM solutions (Table S1), mean 530 groundwater flow lengths and Luxembourg Sandstone thickness of the recharge areas the springs belong to (Table 1), the MRT median values characterizing the response time of spring discharge to effective precipitation input (Table S2), and the average values of hydrochemical parameters (Table S3).

Results are shown in Fig. 12 as a correlation matrix which indicates the sign (blue and red tones for positive and negative correlation, respectively), the strength (the darker the tone, the stronger the correlation and vice versa), and the significance of 535 the linear correlation between each pair of parameters (a black square – $p < 0.05$ – denotes a significant correlation and a white circle – $p < 0.001$ – highlights a very significant one).

It can be observed that $\tau_m$ and $\tau_{exp}$ median values are in general correlated positively with the mineralization of the spring water and negatively with $O_2$ and pH values. However, these correlations are most of the time weak ($|r| < 0.3$) and not significant ($p > 0.05$). Note that the linear correlation of $\tau_{exp}$ median values with $HCO_3^-$ ($r = 0.42$), $Cl^-$ ($r = 0.38$), $Na^+$ ($r = $ 540 0.37) and $SiO_2$ ($r = 0.34$) is nonetheless significant ($p < 0.05$). The relationship relating the mean groundwater flow lengths of

the recharge areas to $\tau_{exp}$ median values is also statistically significant (r = 0.38, p < 0.05), while the one with $\tau_m$ median values is not (r = 0.18, p > 0.05). As shown in Fig. 13a, we can further notice that the correlation between $\tau_{exp}$ median values and the mean groundwater flow lengths at the scale of the recharge areas is characterized by an r value of 0.67.

Finally, the strongest positive and most significant correlations of tritium derived transit times with the other spring characteristics is shown in Fig. 12 for the EPM piston component. Indeed, the correlation of $\tau_{pf}$ median values with the mean Luxembourg Sandstone thickness of the recharge areas and the MRT median values characterizing the response time of spring discharge to effective precipitation input are particularly strong (r = 0.57 and 0.60, respectively) and very significant (p < 0.001). At the scale of recharge areas, the correlation of $\tau_{pf}$ median values with the mean Luxembourg Sandstone's thickness and MRT is characterized by an r value of 0.67 and 0.86, respectively (Fig. 13bc).

## 5. Discussion

The use of LPM for tritium-based water transit time studies of hydrological systems requires several aspects to be considered upfront (e.g., McGuire and McDonnell, 2006; Leray et al., 2016; Stewart and Morgenstern, 2016; Stewart et al., 2017). For instance, the use of a soil water balance model to account for the variation over time of evapotranspiration water losses, combined with the proximity of the Trier Station to our study site, allowed us to assume a suitable characterization of the tritium recharge input signal. The spring samples were also taken at selected times to best consider for the steady-state assumption (although presuming stationarity for groundwater is in general less critical than for stream water), and their analysis was carried out with high-accuracy methods for minimizing the potential impact of tritium analytical errors on modelling results (Stewart et al., 2012; Gallart et al., 2016).

The tritium output data record length is of particularly critical importance to the characterization of hydrological systems in the Northern Hemisphere. While a single tritium sample can be sufficient in the Southern Hemisphere, the use of time series with several measurements remains a prerequisite in the Northern Hemisphere for an unequivocal dating of hydrological systems (Stewart and Morgenstern, 2016). This problem may be tackled by exploiting sites for which past tritium measurements are available (e.g., Stolp et al., 2010; Blavoux et al., 2013; Jerbi et al., 2019). Here, we have explored the potential for the sole use of recent high-accuracy tritium data to date groundwater. We did eventually rely on time series, but of minimal length (as a reminder, time series of 3 samples taken every ~2 years). In this context, we did not use a standard best-fit calibration approach as is most commonly done, but followed the advice of Gallart et al. (2016) to use a modelling framework combining a resampling procedure associated with the GLUE approach (Beven and Binley, 1992, 2014). This approach delivers dating results together with uncertainty ranges reflecting the EPM parameter identifiability issue arising jointly from the low number of spring samples and the tritium analytical errors. Easy to implement and conceptually simple, GLUE is one of the most widely used methods for uncertainty assessment of hydrological models (Moges et al., 2021). However, it is important to note that its subjectivity can be criticized for the choice of the behavioural threshold, the number of sample simulations and their potential impact on the results (Li et al., 2010). Other alternative techniques could have been

used, such as a Bayesian inference framework (e.g., Thiros et al, 2023), but the results would probably have been comparable (e.g., Li et al., 2010; Dotto et al., 2012).

Our findings show that the short spring tritium data time series, even though of high-accuracy, did not allow direct unambiguous dating results. Indeed, even after restricting the mean transit time $\tau_m$ to 35 years maximum to rule out the unlikely oldest solutions considering the hydrogeological context of the study area, the behavioural parameter sets resulting from the random exploration of the focused EPM parameter space still did not define an unambiguous set of solutions, and this for each spring and despite the more stringent NSE threshold of 0.5. This led us to apply an additional 2D kernel NSE

weighted density thresholding filter to conserve the *a priori* most likely solutions and tend towards what we assumed to be the more coherent ensembles of parameter sets. Overall, the NSE figures characterizing these ensembles of most likely solutions (i.e., NSE median values ranging from 0.68 to 0.93 and averaging 0.86) indicate a good match between modelled and observed tritium data. As observed in Fig. S1-S35 and summarized in Fig. 14, the simulated spring tritium concentrations are consistent with the tritium measurements of spring samples, their downward trends, as well as with their analytical errors. Finally, it is

worth noting that a few older tritium data are available for the investigated springs of group KRD (Farlin et al., 2013a, 2013b, 2017). Covering the 2008-2012 period, these data are out of the specific scope of this work because of their lower accuracy (1σ average analytical error of 0.52 TU), hence they were not directly used for the modelling. Included in Fig. S16-S24, we can nonetheless observe that they fit well with the simulation ensembles and therefore represent to some extent an additional consolidating argument for the consistency of the obtained tritium dating results.

Although necessary, the consistency of the results with the tritium measurements is however not sufficient as it does not guarantee their correctness, particularly if the selected model does not adequately characterize the water flow field (Leray et al., 2016; Stewart et al., 2017). Here, we opted for the EPM because of its potential to simulate for unconfined aquifers both the water flowing through the vadose zone (i.e., with the EPM piston segment), as well as the mixing of groundwater flowlines coming from the saturated zone at the outlet of the system (i.e., mimicked by the EPM exponential section; Jurgens et al.,

2012). Even if assessing the validity of the LPM selection is not straightforward (Leray et al., 2016), two key aspects of the findings nevertheless support the EPM choice here. First, the dating results retained *in fine* appear overall physically plausible when translated into equivalent sandstone heights (Fig. 8) using the rational average characteristics of the Luxembourg City spring recharge areas (i.e., outflowing discharge, impluvium, sandstone thickness, porosity). Secondly, the significant and consistent correlations found between $\tau_{pf}$ and $\tau_{exp}$ with several other characteristics describing the vadose and saturated zone

of the Luxembourg Sandstone aquifer (Fig. 12) are also strongly supporting the choice of the EPM model. Indeed, while no straightforward significant correlation of $\tau_m$ values with the other characteristics could be observed, several did appear when $\tau_{pf}$ and $\tau_{exp}$ were regarded separately.

Considering the full thickness of the Luxembourg Sandstone as a proxy for the thickness of its unsaturated zone, the very significant correlation highlighted in Fig. 12 between $\tau_{pf}$ values and the Luxembourg Sandstone thickness means estimates of

605 each group of springs is a first element that supports the ability of the EPM piston segment to deliver plausible transit times

characterizing the percolation of water through the vadose zone of the aquifer. Although the thickness of the unsaturated zone could not be directly estimated, we conjecture that it is positively correlated with the Luxembourg Sandstone thickness, assuming that the saturated zone thickness is comparable from one area to another (which is reasonable given the perched character of the aquifer and the gently subhorizontal sloping topography of its base characterizing each hydrogeological district). The Luxembourg Sandstone thickness $-\tau_{pf}$ relationship presented at the scale of the recharge area in Fig. 13b indicates, however, that group KRD springs deviate overall from the general behaviour with particularly high $\tau_{pf}$ values. A possible explanation for this specific behaviour is the known presence of a marly layer intercalated in the Luxembourg Sandstone formation in the KRD area (as shown by the observational borehole drilled there; Farlin et al., 2013a). A second argument that supports the EPM piston component's potential for adequately reproducing the transit times required for infiltrating water to cross the vadose zone is the highly significant correlation of $\tau_{pf}$ values with response times of spring discharges deduced from the cross-correlation analysis (i.e., MRT values; Fig. 12 and Fig. 13c). Such cross-correlation time series analysis of response times to effective precipitation inputs have indeed been related to the characteristics of the unsaturated zone for several other unconfined aquifers (e.g. Lee and Lee, 2000; Lee et al., 2006; Fiorillo and Doglioni, 2010; Bloomfield and Marchant,2013; Bloomfield et al.,2015; Al-Jaf et al., 2021). As shown in Fig. 15, it is worth noting that the MRT - $\tau_{pf}$ relationship seems linear, but not one-to-one. MRT values are in general almost four times lower than $\tau_{pf}$ ones. A potential explanation for this difference is that MRT would reflect water celerity, i.e., the speed at which the water infiltration signal is transmitted through the vadose zone to the springs, while $\tau_{pf}$ relate to water velocity, i.e., the movement of water molecules (McDonnell and Beven, 2014; Scaini et al., 2017; van Verseveld et al., 2017; Shao et al., 2018; Worthington and Foley, 2021).

Although none of the studied physicochemical parameters were highlighted as a direct reliable proxy of water transit times in the Luxembourg Sandstone aquifer, the global correlation pattern of $\tau_{exp}$ with the hydrochemistry of the spring waters (Fig. 12) suggests an overall progressive mineralization of groundwater with increasing flow path length in the saturated zone. The significant correlation between $\tau_{exp}$ values and the average groundwater flow lengths estimated for each of the recharge areas also allows the exponential component of the EPM model to be related to the saturated zone of the aquifer (Fig. 12). It should nevertheless be noticed that the average groundwater flow length characterizing the recharge area of group B seems overall relatively short compared to the relatively large $\tau_{exp}$ values that have been calculated for the springs that drain it (Fig. 13a). This inconsistency, which has already been raised indirectly when translating the tritium dating results into equivalent sandstone heights (see 4.1.3 and Fig. 8), could however be due to an incorrect estimate of the impluvium of this group of springs and would therefore not reflect a problem with the tritium-EPM derived dating results. It is moreover worth noting that a previous study (Luxconsult, 1992) estimated the recharge area of Group B springs (shown in Gourdol et al., 2013) to be twice as large as the one newly defined (RGD, 2021b) – a discrepancy that we have not yet been able to explore further.

To conclude on the reliability of the tritium dating results obtained in this study, a rough assessment of water velocities in the vadose and saturated zones of the aquifer can be undertaken. First, we can assume that the EPM piston and exponential

components reflect well the water flowing through the vadose zone and the mixing of groundwater flowlines coming from the saturated zone at the outlet. Second, by dividing $\tau_{pf}$ and $\tau_{exp}$ values by the mean values of the vadose zone thickness and the groundwater flow length in the saturated zone respectively, we can approximately estimate water velocities in these two specific compartments of the Luxembourg Sandstone aquifer. Conducted at the scale of the recharge area (using the average of $\tau_m$ and $F$ median values (Table S1) of each group of springs, mean groundwater flow lengths and Luxembourg Sandstone thicknesses reported in Table 1, and $H_{exp}$ values as defined in section 4.1.3), this assessment leads to very close figures from one area to another (excluding group B results, whose impluvium was previously questioned). Ranging from 1.9 x 10-7 to 5.5 x 10-7 m/s, the water velocity determined for the vadose zone is on average 3.7 x 10-7 m/s (~12 m/year). About an order of magnitude higher in all cases, the water velocity assessed for the saturated zone ranges from 4.0 x 10-6 to 8.7 x 10-6 m/s and reaches 5.4 x 10-6 m/s on average (~170 m/year). This tritium derived anisotropy of water velocity shared by all the studied hydrogeological areas is consistent with the well-known anisotropy of hydraulic properties characterizing bedded aquifers introduced in section 2.1. Several studies focusing on the characterization of the permeability anisotropy have in particular shown that horizontal permeability of bedded sandstone formations is greater than their vertical permeability, and this up to several orders of magnitude (e.g., Hitchmough et al., 2007; Clavaud et al., 2008; Goupil et al., 2022).

To the best of our knowledge, even if observed by Fox et al. (2008), the anisotropy of the Luxembourg Sandstone permeability has never been quantified and its characterization may require further attention in future studies. This kind of data, associated with other existing datasets (e.g., Meus and Willems, 2021), could then be used to feed a physically-based 4D model (e.g., Paradis et al., 2018) that would more accurately represent the multi-scale complexity of the Luxembourg Sandstone bedrock aquifer (Worthington, 2015). The comparison of the output from such a model with that resulting from this study would allow the predictive capacities to be assessed – for instance for pollutant contamination mitigation purposes – of the tritium based LPM approach used here (Eberts et al., 2012; Yager et al., 2013; Marçais et al., 2015; Balvin et al., 2021). Potential sources of error that remained unaddressed in this study, such as aggregation biases (Bethke and Johnson, 2008; Stewart et al., 2017) resulting from a multi-modal mixture of waters of different ages associated with the properties of the fissured-porous aquifer (e.g., Rajaram, 2021), could indeed be explored in this way. Additionally, employing environmental tracers typically used for dating old groundwaters could further aid in mitigating these concerns (Bethke and Johnson, 2008; Jasechko, 2016; Rädle et al., 2022).

## 6. Conclusion

We have presented a new set of 105 high-accuracy tritium analyses, carried out on water samples collected three times at intervals of ~2 years for 35 springs, each of them fed by the Luxembourg Sandstone aquifer. We leveraged these short but high-accuracy tritium content time series to investigate the potential of the tritium dating technique to determine young groundwater transit times in the Northern Hemisphere at new study sites where no historical tritium data is available.

Groundwater mean transit times have been assessed using the LPM approach in a modelling framework, delivering uncertainty ranges inherent to the low number of tritium data at hand and their analytical errors.

Our results show first that unambiguous groundwater mean transit time estimations solely based on recent short tritium content time series are not possible, as several ranges of mean transit times appeared plausible. Nonetheless, we succeeded through a stepwise decision process to discriminate for every spring an ensemble of most likely solutions, consistent with both the tritium

measurements and the overall hydrogeological context of our area of interest. This assessment was possible because several independent supplementary data was available for each spring, and because of the large number of investigated springs and their varying characteristics.

Our work has demonstrated the suitability of the EPM model to assess groundwater mean transit times (through the vadose and saturated zones of the Luxembourg Sandstone aquifer) from high-accuracy tritium analyses of spring water samples. When

translated into water velocities, the obtained dating results appeared particularly consistent with the horizontal-vertical anisotropy of the aquifer's hydraulic properties due to its bedded character – a key feature that deserves further investigations. To conclude, this work not only advanced our knowledge of water transit times in the Luxembourg Sandstone aquifer but should also be considered as an example of how dating of young groundwater bodies at new sites is today feasible with short tritium data series in Central Europe.

**Code and data availability**

The R (R Core Team, 2022) scripts and data used to achieve this study are available at https://doi.org/10.5281/zenodo.11506262 (Gourdol et al., 2024).

**Author contributions**

LG conceptualized the study with guidance from MKS, UM and LP. LG implemented the field and modelling work, conducted

the data analysis and wrote the first draft of the paper. All the co-authors (LG, MKS, UM, and LP) contributed to and edited the manuscript.

**Competing interests**

A co-author is a member of the editorial board of Hydrology and Earth System Sciences journal. The peer-review process was guided by an independent editor, and the authors have also no other competing interests to declare.

**Acknowledgements**

This research work spined of a long-term public–public collaboration between the LIST and the City of Luxembourg. We especially thank the Water Service of Luxembourg City for their continuous and highly valuable support. We thank the LIST colleagues Cyrille Tailliez and Jerôme Juilleret for their help during the field work, François Barnich for the hydrochemical analyses, Renaud Hostache and Davide Zoccatelli for fruitful discussions about the modelling aspect of this work, and the technical staff of the GNS Tritium & Water Dating Laboratory for the tritium analyses. The authors are deeply grateful to Francesc Gallart for providing access to the code of the modelling framework implemented in Gallart et al. (2016). We would also like to thank the anonymous reviewers whose comments helped to improve the manuscript.

**Financial support**

The spring discharge and hydrochemical data used in this study was jointly funded by the LIST and the City of Luxembourg, the spring tritium data was funded by the LIST.

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

**Table 1.** Recharge area attributes.

| Recharge area | Area (km²) | Mean discharge (m³/day) | Mean elevation (m) | Mean flow length estimation (m) | Mean Luxembourg Sandstone thickness (m) |
|---|---|---|---|---|---|
| B | 4.00 | 2900 | 363.9 | 1363 | 18.2 |
| D | 0.83 | 400 | 373.6 | 951 | 48.8 |
| G | 7.18 | 4500 | 376.5 | 1261 | 36.5 |
| KRD | 3.89 | 2700 | 341.7 | 944 | 43.5 |
| KRG | 3.01 | 2900 | 347.3 | 1190 | 68.1 |
| M | 2.01 | 800 | 340.7 | 989 | 35.6 |
| S | 7.17 | 5000 | 335.5 | 1896 | 49.3 |
| P | 4.41 | 3000 | 327.6 | 2388 | 49.6 |

**Table 2.** Tritium analytical results for the investigated springs of Luxembourg City.

| Spring | Recharge area | 10/06/2011 $^3$H ± 1σ (TU ± TU) | 06/09/2013 $^3$H ± 1σ (TU ± TU) | 17/08/2015 $^3$H ± 1σ (TU ± TU) | 18/08/2015 $^3$H ± 1σ (TU ± TU) | 21/08/2017 $^3$H ± 1σ (TU ± TU) | 22/08/2017 $^3$H ± 1σ (TU ± TU) |
|---|---|---|---|---|---|---|---|
| B01 | B | - | 6.624 ± 0.094 | 6.098 ± 0.102 | - | 5.550 ± 0.090 | - |
| B02 | B | - | 6.092 ± 0.101 | 5.627 ± 0.095 | - | 5.177 ± 0.082 | - |
| B03 | B | - | 6.382 ± 0.095 | 5.702 ± 0.093 | - | 5.391 ± 0.083 | - |
| B06 | B | - | 6.321 ± 0.104 | 5.582 ± 0.091 | - | 5.100 ± 0.088 | - |
| B07 | B | - | 6.281 ± 0.104 | 5.879 ± 0.094 | - | 5.568 ± 0.075 | - |
| B09 | B | - | 6.575 ± 0.116 | 5.791 ± 0.094 | - | 5.751 ± 0.079 | - |
| B10 | B | - | 6.383 ± 0.094 | 5.759 ± 0.093 | - | 5.366 ± 0.092 | - |
| C01 | G | - | 7.124 ± 0.097 | 6.196 ± 0.100 | - | 5.741 ± 0.090 | - |
| C03 | G | - | 6.322 ± 0.110 | 5.436 ± 0.096 | - | 4.991 ± 0.087 | - |
| C04 | G | - | 6.005 ± 0.087 | 5.329 ± 0.087 | - | 5.017 ± 0.079 | - |
| C05 | G | - | 5.579 ± 0.066 | 5.416 ± 0.089 | - | 4.960 ± 0.080 | - |
| C07 | G | - | 6.153 ± 0.094 | 5.458 ± 0.090 | - | 5.070 ± 0.090 | - |
| C09 | G | - | 5.965 ± 0.101 | 5.360 ± 0.088 | - | 5.080 ± 0.080 | - |
| C10 | G | - | 5.819 ± 0.104 | 5.275 ± 0.093 | - | 4.900 ± 0.090 | - |
| D01 | D | - | 5.928 ± 0.085 | 5.269 ± 0.096 | - | 5.020 ± 0.090 | - |
| K01 | KRD | - | 5.426 ± 0.084 | - | 5.162 ± 0.095 | - | 4.847 ± 0.088 |
| K02 | KRD | - | 5.277 ± 0.090 | - | 4.859 ± 0.091 | - | 4.684 ± 0.087 |
| K03 | KRD | - | 5.410 ± 0.097 | - | 5.212 ± 0.092 | - | 4.589 ± 0.085 |
| K07 | KRD | - | 5.671 ± 0.081 | - | 4.906 ± 0.091 | - | 4.618 ± 0.085 |
| K13 | KRD | - | 5.812 ± 0.083 | - | 4.908 ± 0.091 | - | 4.592 ± 0.085 |
| K17 | KRD | - | 5.479 ± 0.095 | - | 4.873 ± 0.092 | - | 4.429 ± 0.082 |
| K19 | KRD | - | 5.732 ± 0.099 | - | 5.071 ± 0.095 | - | 4.711 ± 0.087 |
| K21 | KRD | 7.125 ± 0.090 | - | - | 5.175 ± 0.097 | - | 4.681 ± 0.086 |
| K21A | KRD | 6.080 ± 0.090 | - | - | 4.695 ± 0.089 | - | 4.310 ± 0.081 |
| K22 | KRG | - | 5.694 ± 0.082 | - | 5.067 ± 0.095 | - | 4.719 ± 0.080 |
| K24 | KRG | - | 5.405 ± 0.087 | - | 4.987 ± 0.094 | - | 4.763 ± 0.072 |
| K26 | KRG | - | 5.508 ± 0.096 | - | 4.928 ± 0.082 | - | 4.708 ± 0.090 |
| K28 | KRG | - | 5.587 ± 0.097 | - | 5.033 ± 0.084 | - | 4.700 ± 0.085 |
| K31 | KRG | - | 5.724 ± 0.081 | - | 5.051 ± 0.085 | - | 4.738 ± 0.084 |
| K32 | KRG | - | 5.570 ± 0.082 | - | 5.158 ± 0.086 | - | 4.591 ± 0.084 |
| M01 | M | - | 5.794 ± 0.100 | 5.462 ± 0.099 | - | 4.960 ± 0.090 | - |
| P01 | P | - | 6.729 ± 0.113 | 5.843 ± 0.104 | - | 5.330 ± 0.090 | - |
| S01 | S | - | 5.763 ± 0.099 | 5.302 ± 0.098 | - | 4.870 ± 0.080 | - |
| S02 | S | - | 6.098 ± 0.107 | 5.452 ± 0.096 | - | 4.810 ± 0.080 | - |
| S03 | S | - | 5.953 ± 0.102 | 5.342 ± 0.097 | - | 4.670 ± 0.080 | - |

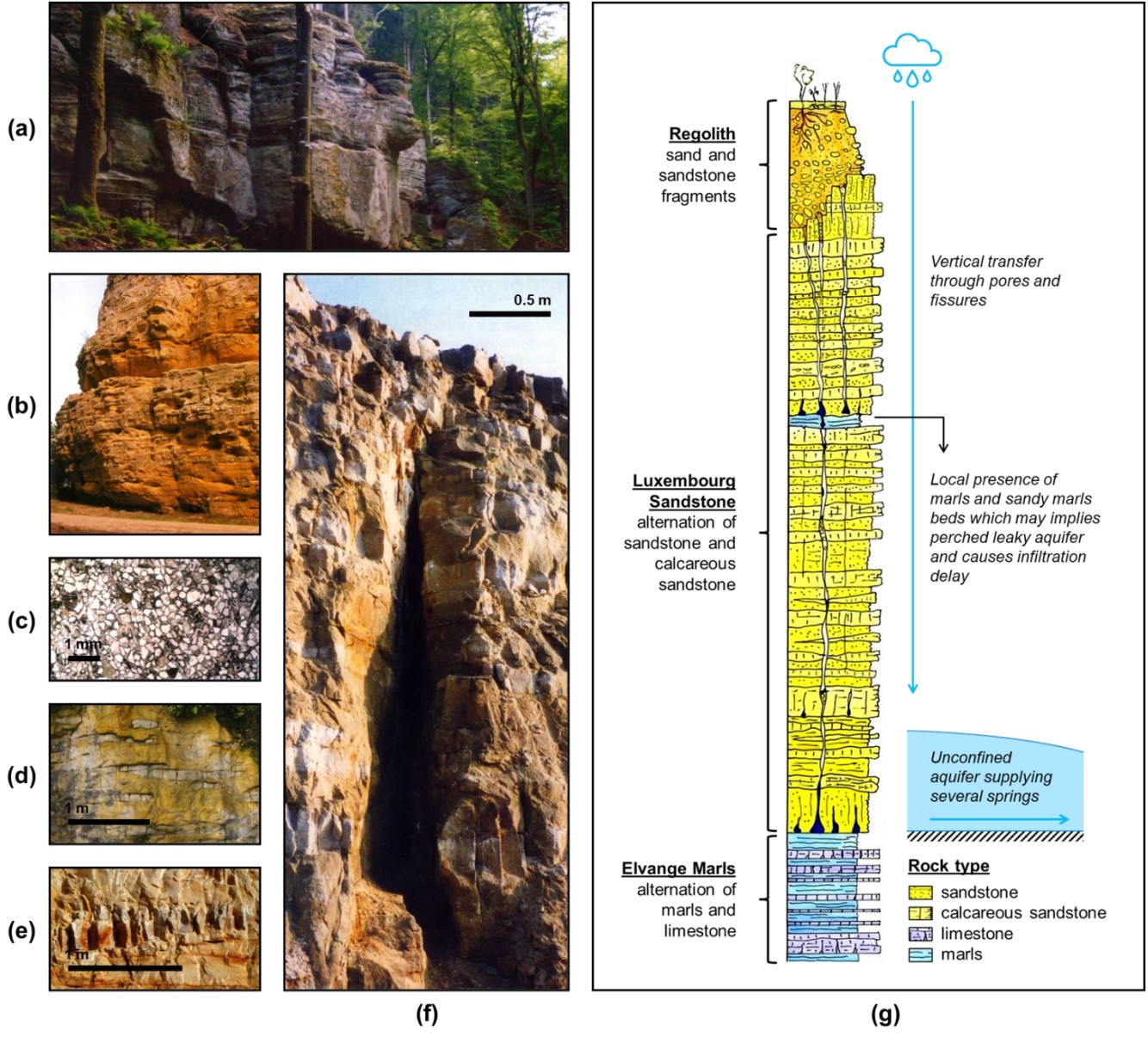

**Figure 1.** Overview of the Luxembourg Sandstone (a-f) in pictures (modified from the proceedings of the 2nd international conference on sandstone landscapes, Vianden, Luxembourg, 25-28/05/2005 (Ries and Krippel, 2005) with (a) sandstone outcrop in the Mullerthal area (photo taken by Jiri Adamovic during the pre-conference excursion), (b) sandstone outcrop in the Pétrusse valley in Luxembourg City (Faber and Weis, 2005), (c) optical microscope image of a well cemented sandstone sample under natural light (Colbach, 2005), (d) outcrop showing an alternation of carbonate rich, whitish, sandy limestone and carbonate poor, yellowish sandstone (Colbach, 2005), (e) fresh roadcut showing a closely fractured carbonate-rich bed (Colbach, 2005), and (f) roadcut showing a wide open fracture (Colbach, 2005)) and (g) through a typical lithological log (adapted from a sketch drawn in 2001 by Robert Maquil from *Administration des ponts et chaussées, Service géologique de l'État, Luxembourg*).

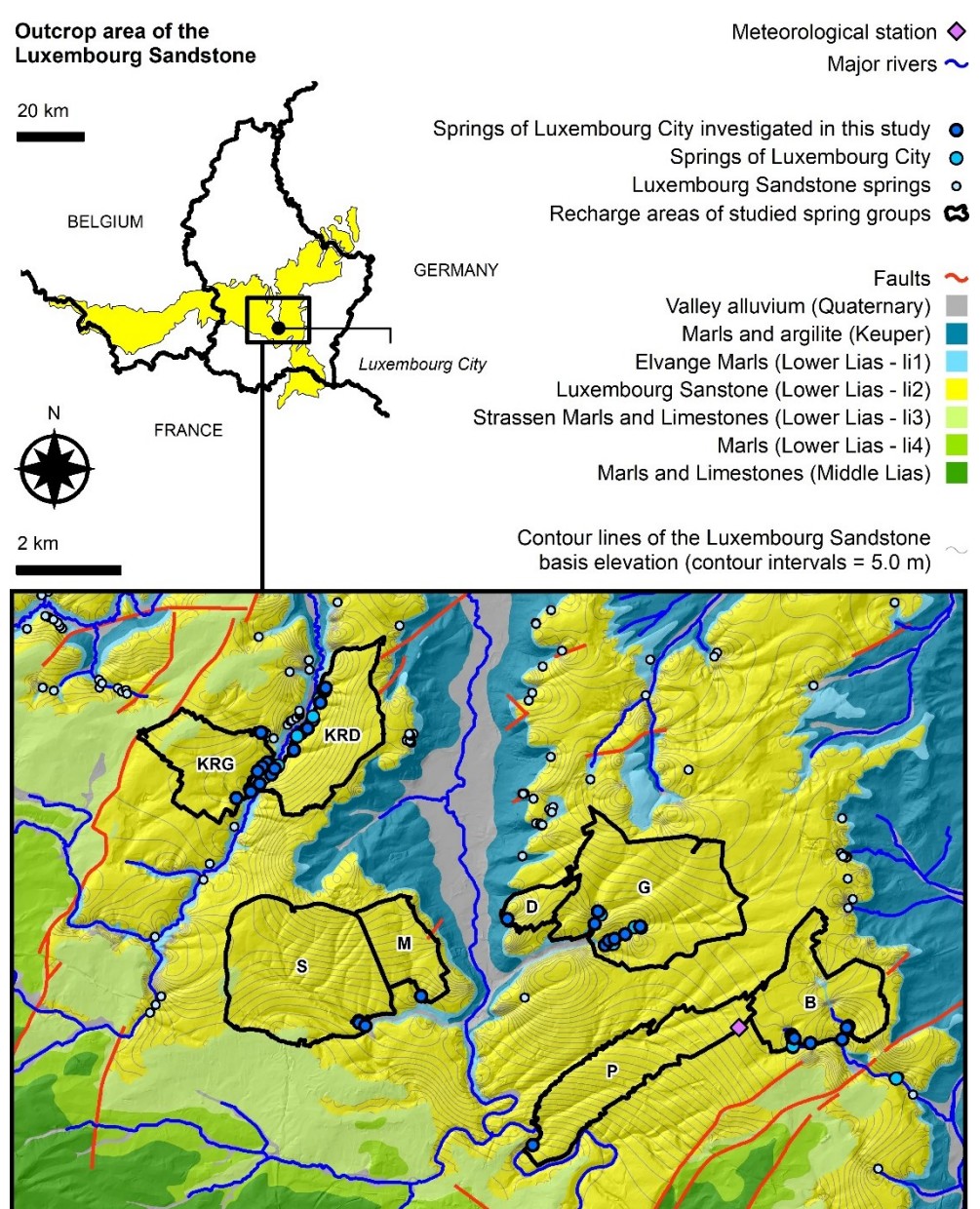


**Figure 2.** Geological and hydrogeological settings of the study area (the geological units delineation comes from the Luxembourg harmonized geological map from *Administration des ponts et chaussées, Service géologique de l'État, Luxembourg*; the springs of Luxembourg City are contact springs located at the boundary between the Luxembourg Sandstone and the underlying Elvange Marls; the 35 springs of Luxembourg City investigated in this study are B01, B02, B03, B06, B07, B09, B10 for group B, C01, C03, C04, C05, C07, C09, C10 for group G, D01 for group D, K01, K02, K03, K07, K13, K17, K19, K21, K21A for group KRD, K22, K24, K26, K28, K31, K32 for group KRG, M01 for group M, P01 for group P, and S01, S02, S03 for group S).


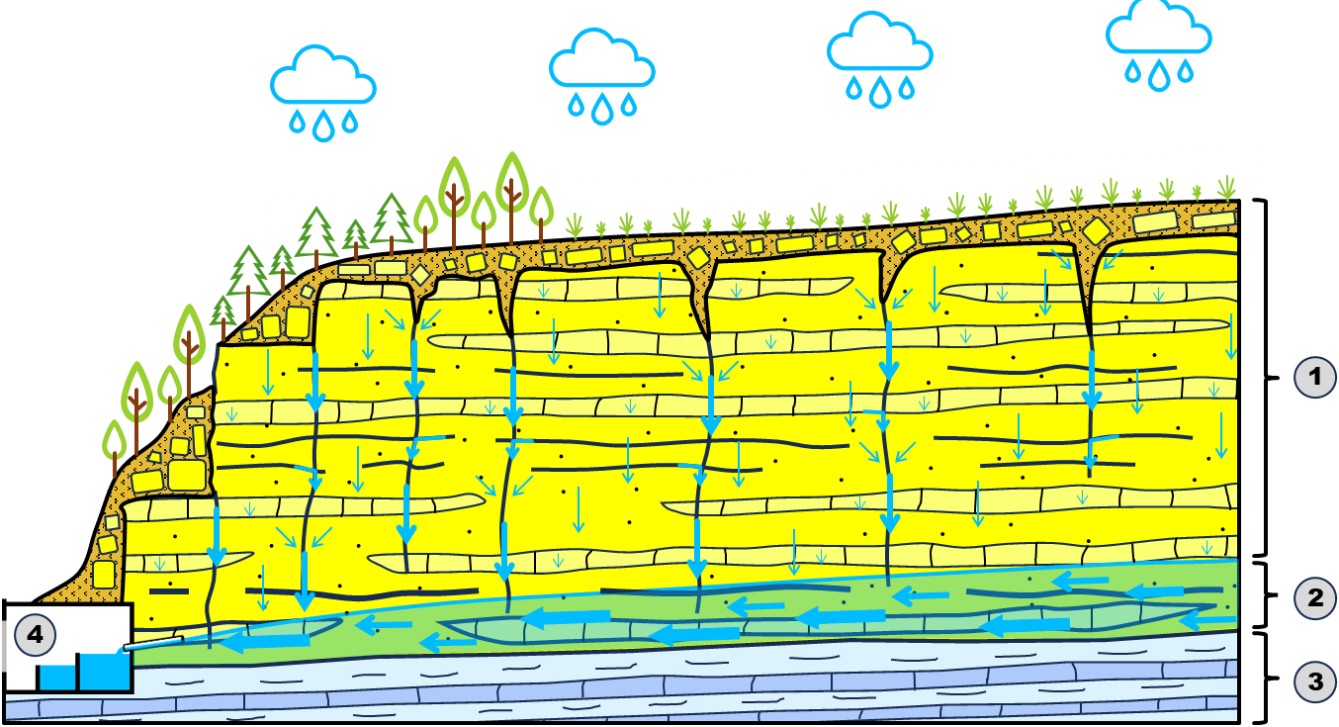

**Figure 3.** Synthetic geological cross-section illustrating the catchment facility of a spring draining the Luxembourg Sandstone aquifer and the water flow components within the bedrock (the thickness of arrows is proportional to flow velocity; 1: slow and fast vertical transfer of water in the unsaturated zone trough pores and fractures, respectively; 2: subhorizontal water flows in the saturated zone with fastest flows in main fractures; 3: Elvange Marls impervious unit constituting the base of the aquifer; 4: spring catchment facility; one can note that Strassen Marls and Limestones are absent in this case; adapted from Meus and Willems, 2021).


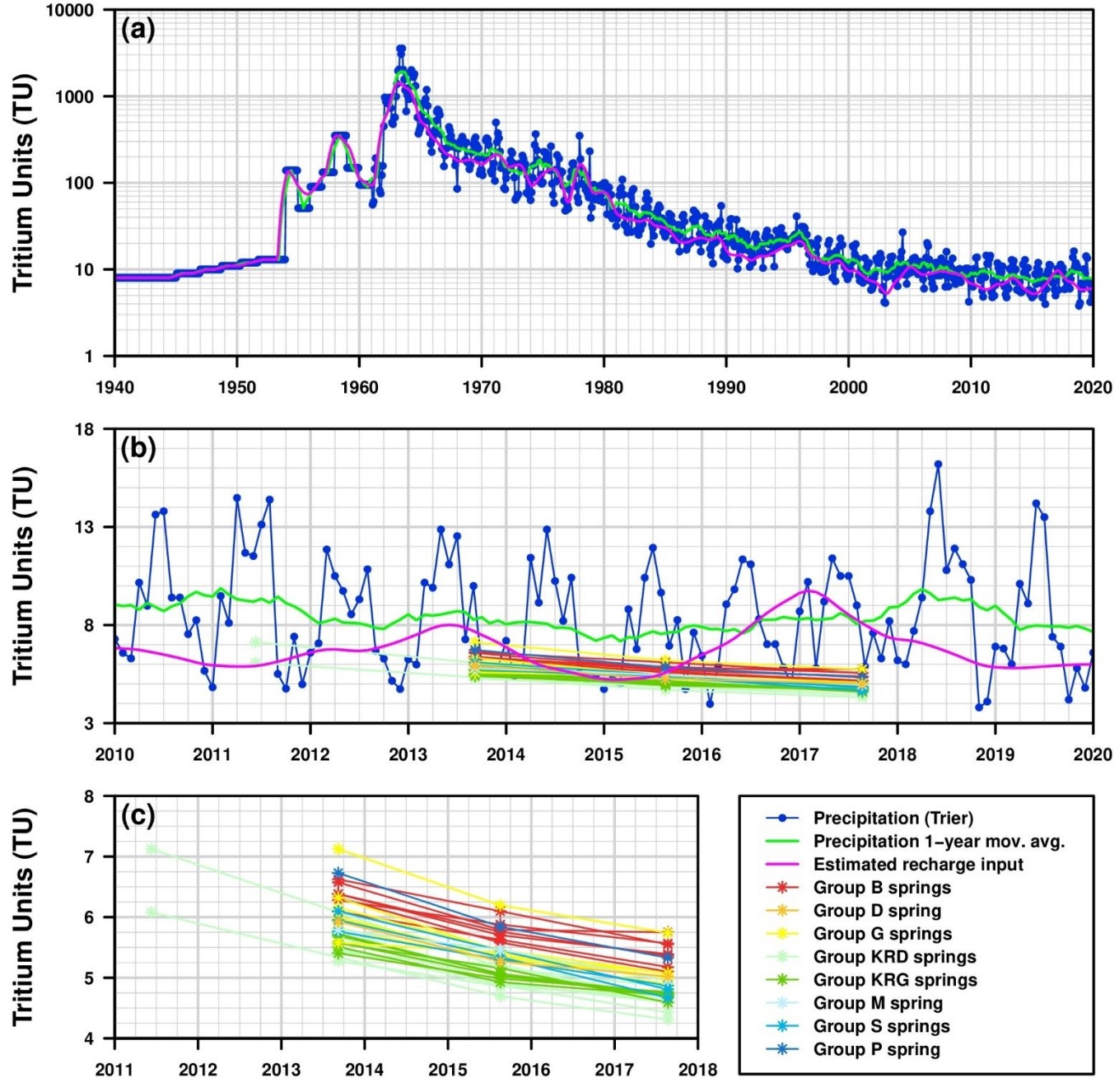

**Figure 4.** Tritium concentration in (a, b) precipitation, recharge, and (b, c) investigated springs of Luxembourg City (a) from 1940 to 2020 with a log scale, and (b) from 2010 to 2020 and (c) 2011 to 2018 with a linear scale.


**Figure 5**

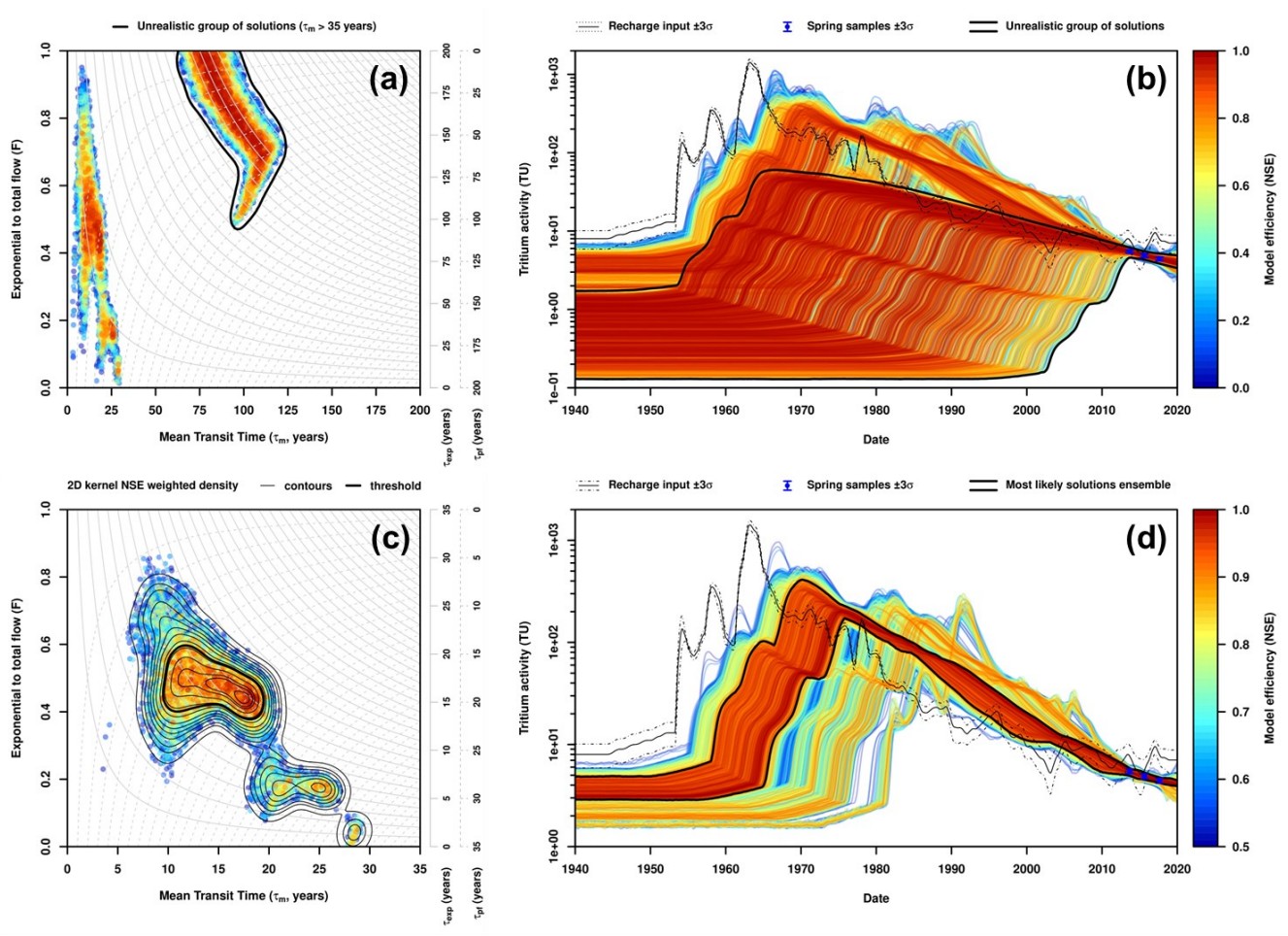

**Figure 5.** Modelling results of spring K17 in the EPM parameter space (a, c) and associated modelled tritium time series (b, d) for the first (a, b) and the second (c, d) screening stages.

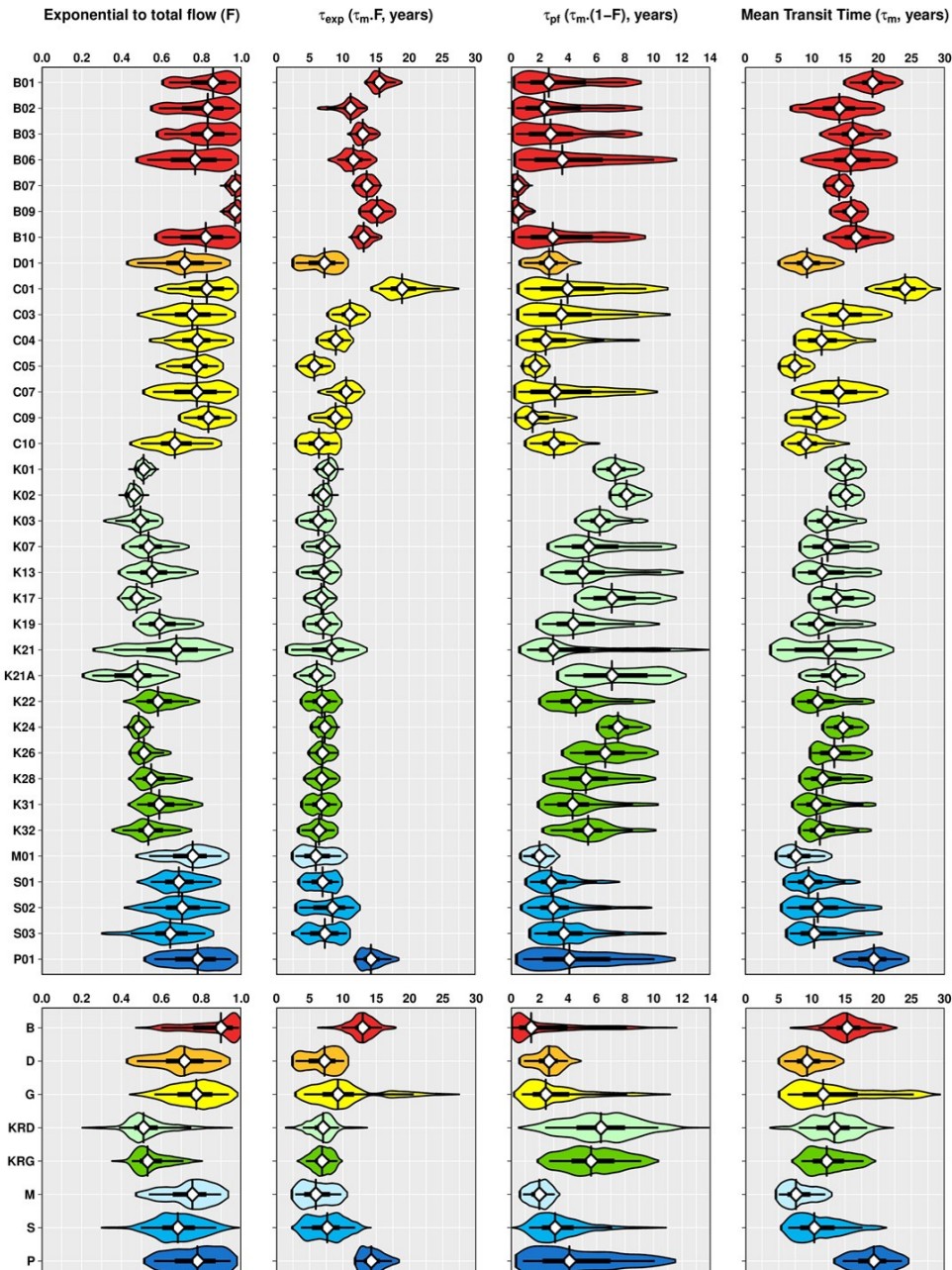

**Figure 6.** Violin plot with boxplot overlay representation of the ensembles of most likely EPM model solutions retained for each spring (upper panels) and by group of springs from the same recharge area (lower panels). The violin plot lines correspond to a kernel density of the distribution (i.e., a smoothed histogram). The violin plot fill colors allow to discriminate the different groups of springs. The white diamond shape with the vertical line indicates the median of the distribution. The horizontal thick and thin lines represent the interquartile and 5-95th percentile ranges of the distribution, respectively.


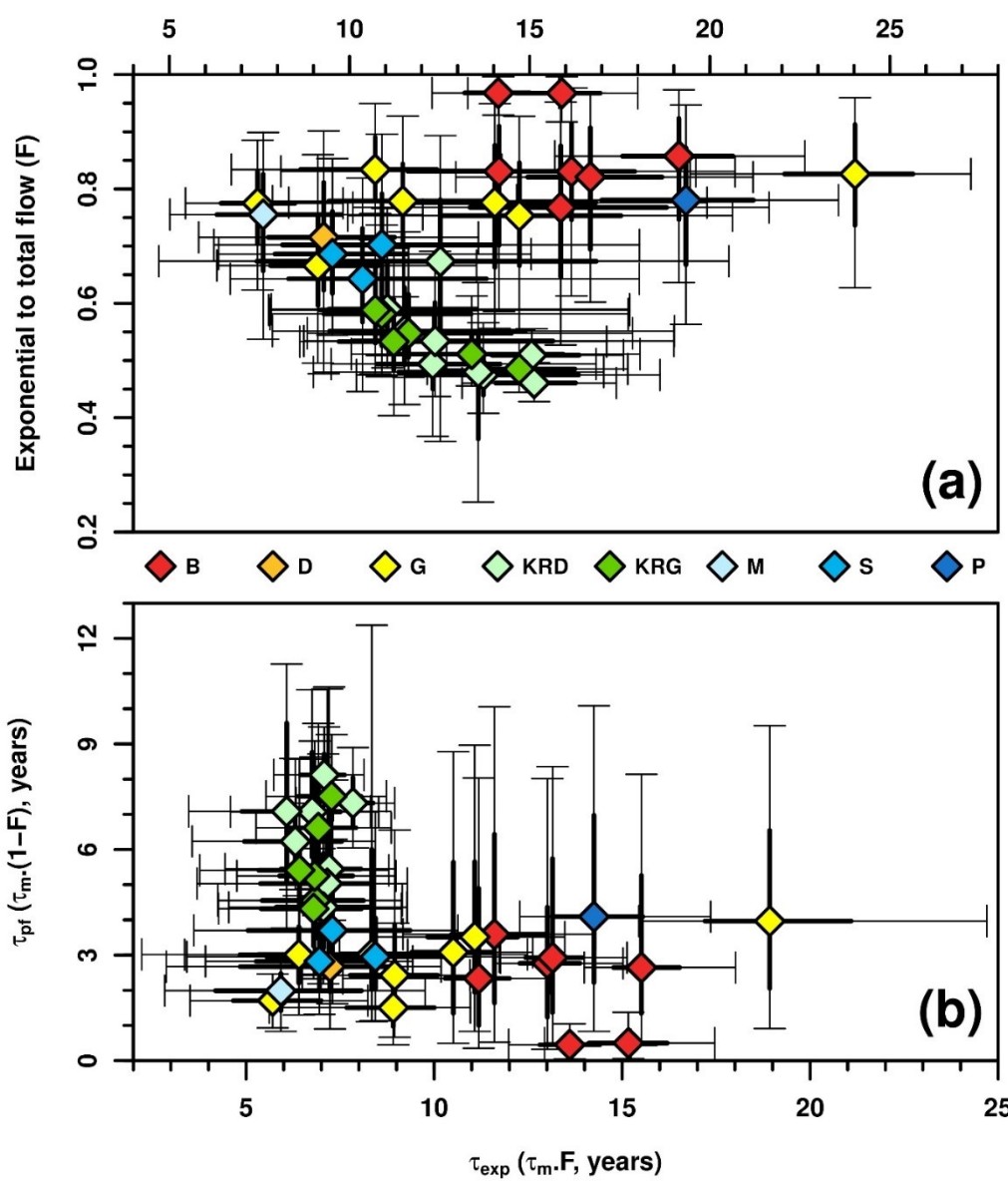


**Figure 7.** Scatter plots relating (a) $F$ to $\tau_m$ and (b) $\tau_{pf}$ to $\tau_{exp}$ values of the ensembles of most likely EPM model solutions retained for each spring. The diamond shape indicates the median of the distribution. The diamond shape fill colors allow to discriminate the different groups of springs. The thick lines and bounded thin lines represent the interquartile and 5-95th percentile ranges of the distribution, respectively.


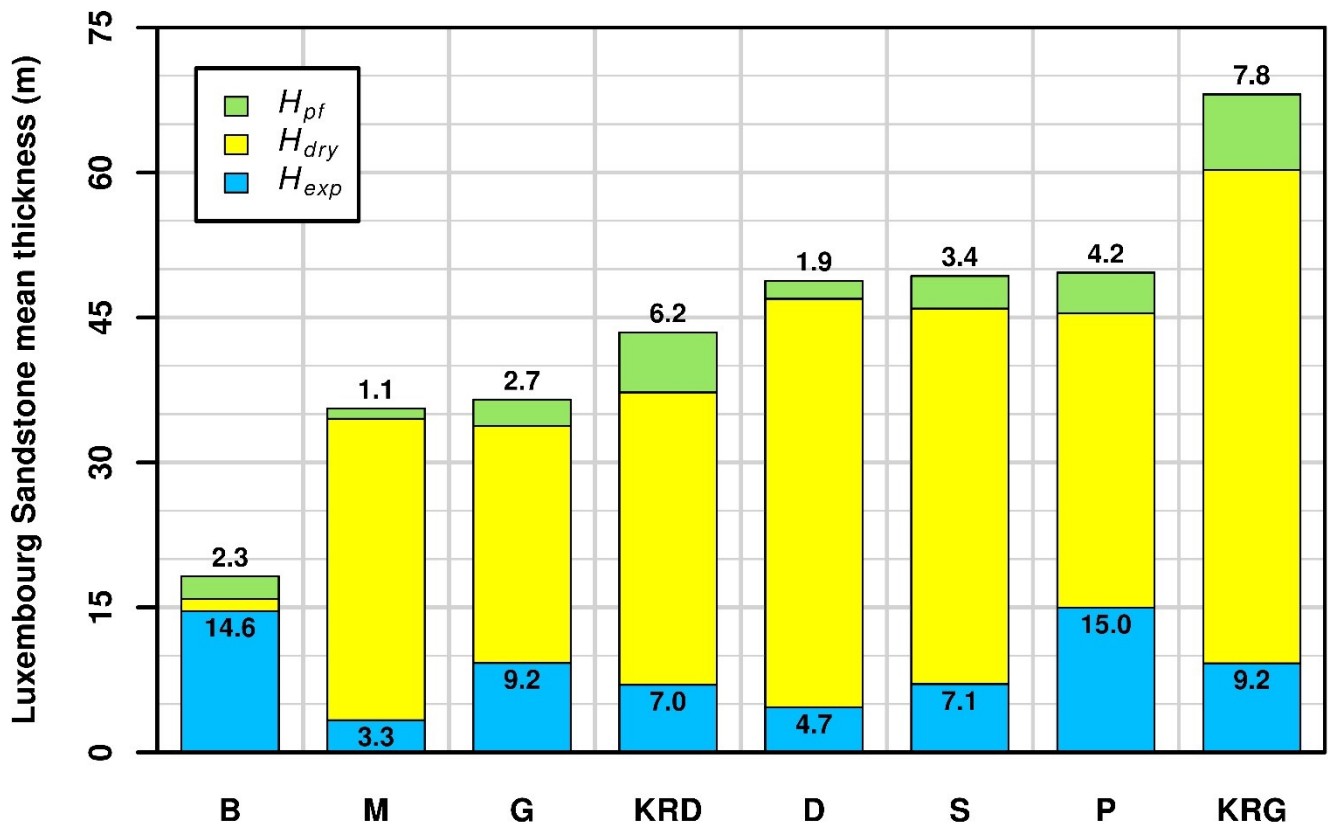

**Figure 8.** Transit time derived mean sandstone layers required for the water storage in the saturated zone (assumed to match the exponential part of the EPM model; $H_{exp}$) and the vadose zone (assumed to correspond to the EPM piston segment; $H_{pf}$) overlaid to the mean thickness of the Luxembourg Sandstone estimated for each recharge areas. $H_{dry}$ stands for the mean sandstone equivalent dry layer and is equal to the mean thickness of the Luxembourg Sandstone minus the sum of $H_{pf}$ and $H_{exp}$ values for each recharge areas.

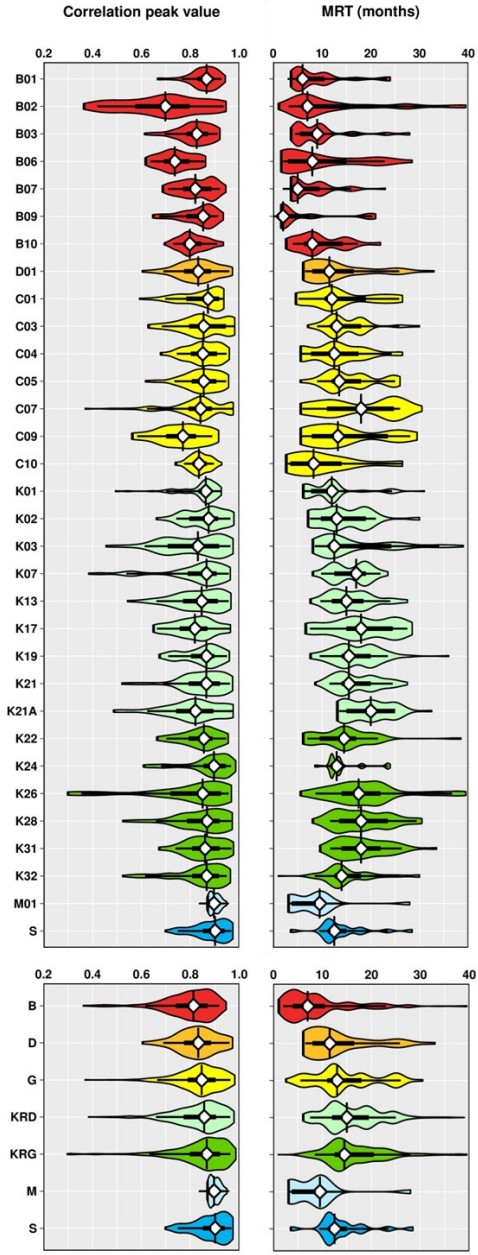


**Figure 9.** Violin plot with boxplot overlay representation of the ensembles of solutions resulting from the spring discharge – effective precipitation cross-correlation analysis for each spring (upper panels) and by group of springs from the same recharge area (lower panels). The violin plot lines correspond to a kernel density of the distribution (i.e., a smoothed histogram). The violin plot fill colors allow to discriminate the different groups of springs. The white diamond shape with the vertical line indicates the median of the distribution. The

horizontal thick and thin lines represent the interquartile and 5-95th percentile ranges of the distribution, respectively.

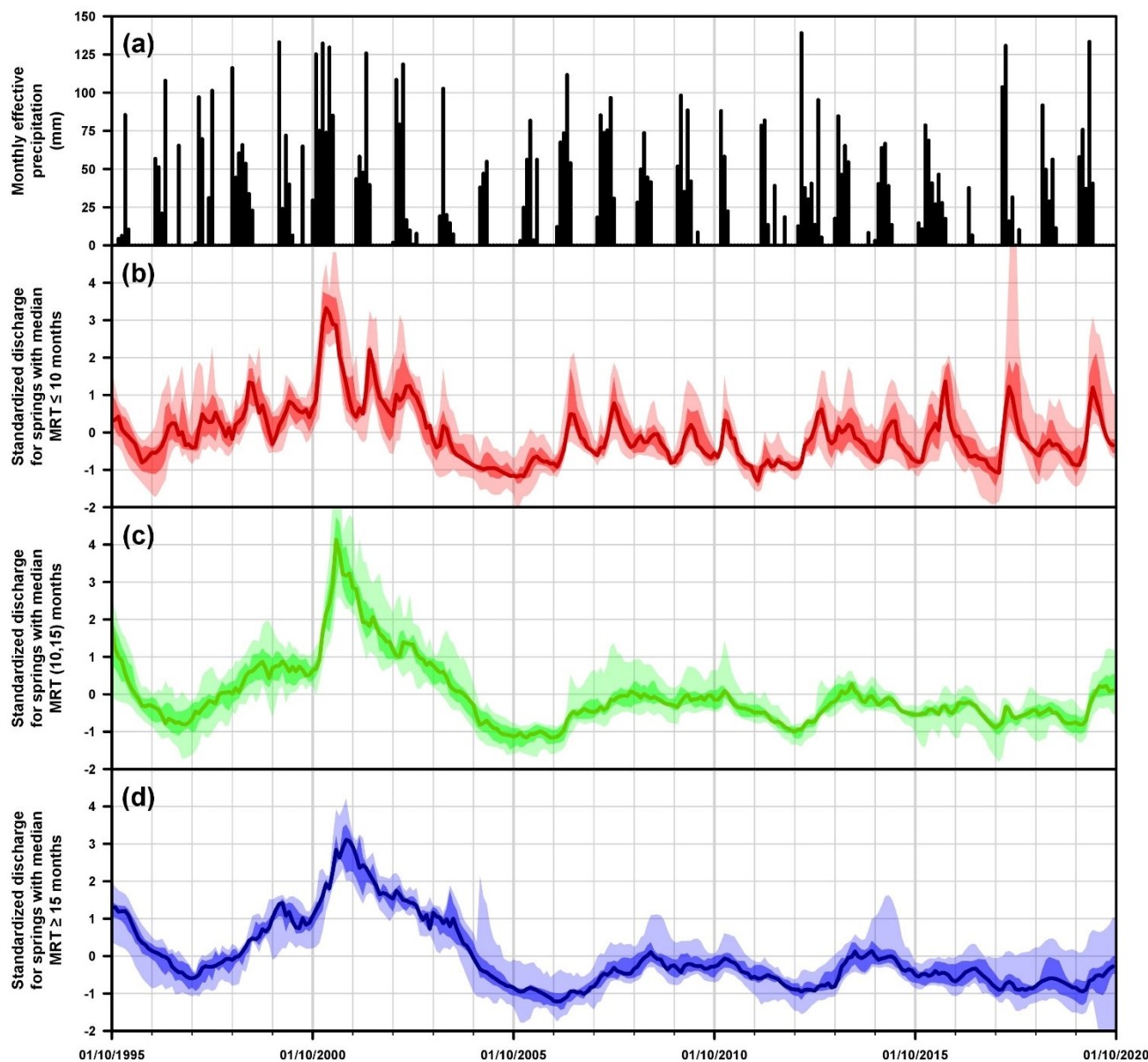

**Figure 10.** Effective precipitation and spring discharge dynamics with (a) monthly effective precipitation, and (b) standardized discharge time series for springs with a median MRT less than or equal to 10 months, (c) in between 10 and 15 months and (d) greater than or equal to 15 months. In (b-d) the colored thick solid line correspond to the median value of the ensemble of spring discharge time series, while darker and lighter colored shaded area correspond to their interquartile and 5-95th percentile envelopes, respectively.



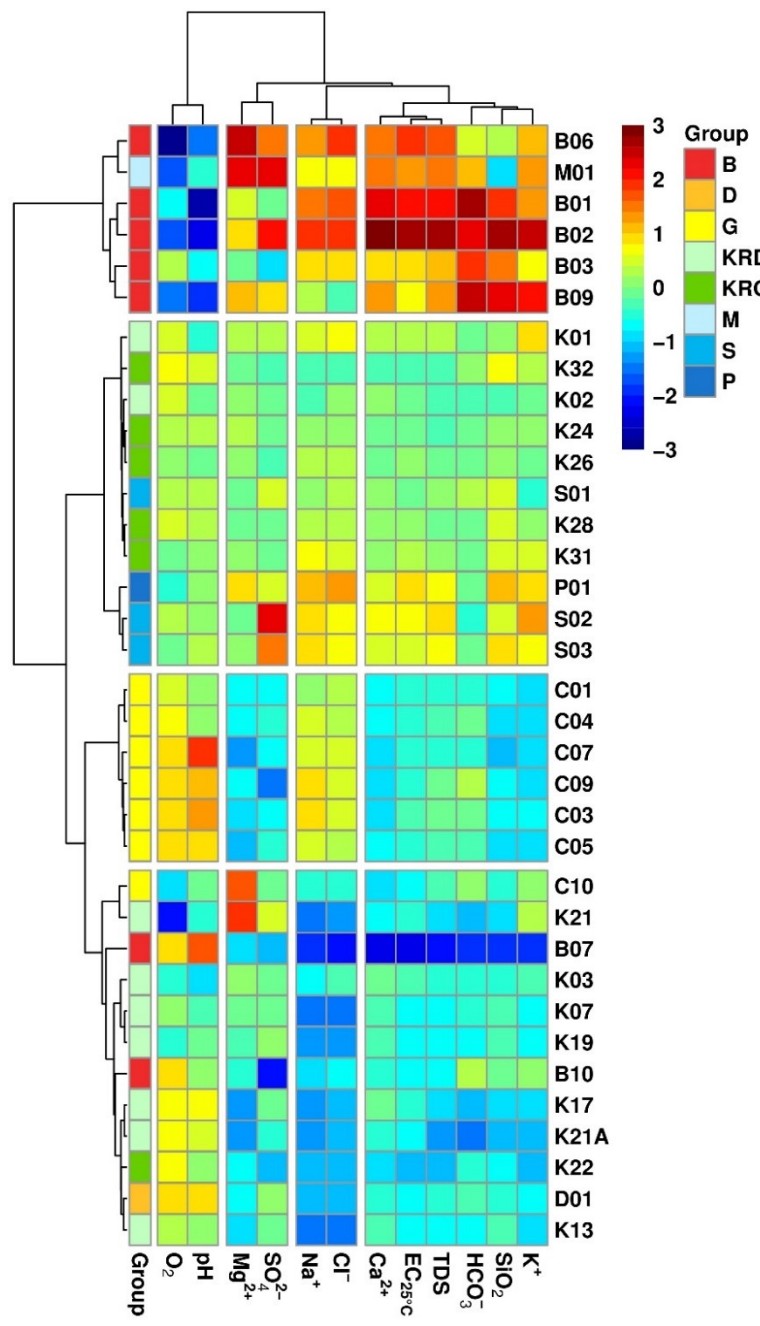

**Figure 11.** Heat map of the average physicochemical characteristics for the investigated springs. Springs and parameters were classified and grouped using a hierarchical cluster analysis (distance measure: Euclidean distance, linkage rule: Ward's method). Prior to the analysis, data were first log-transformed and then each parameter was standardized. According to the dendrograms and the visual inspection of the heat map, 4 clusters of springs and 4 clusters of parameters were emphasized in the figure.


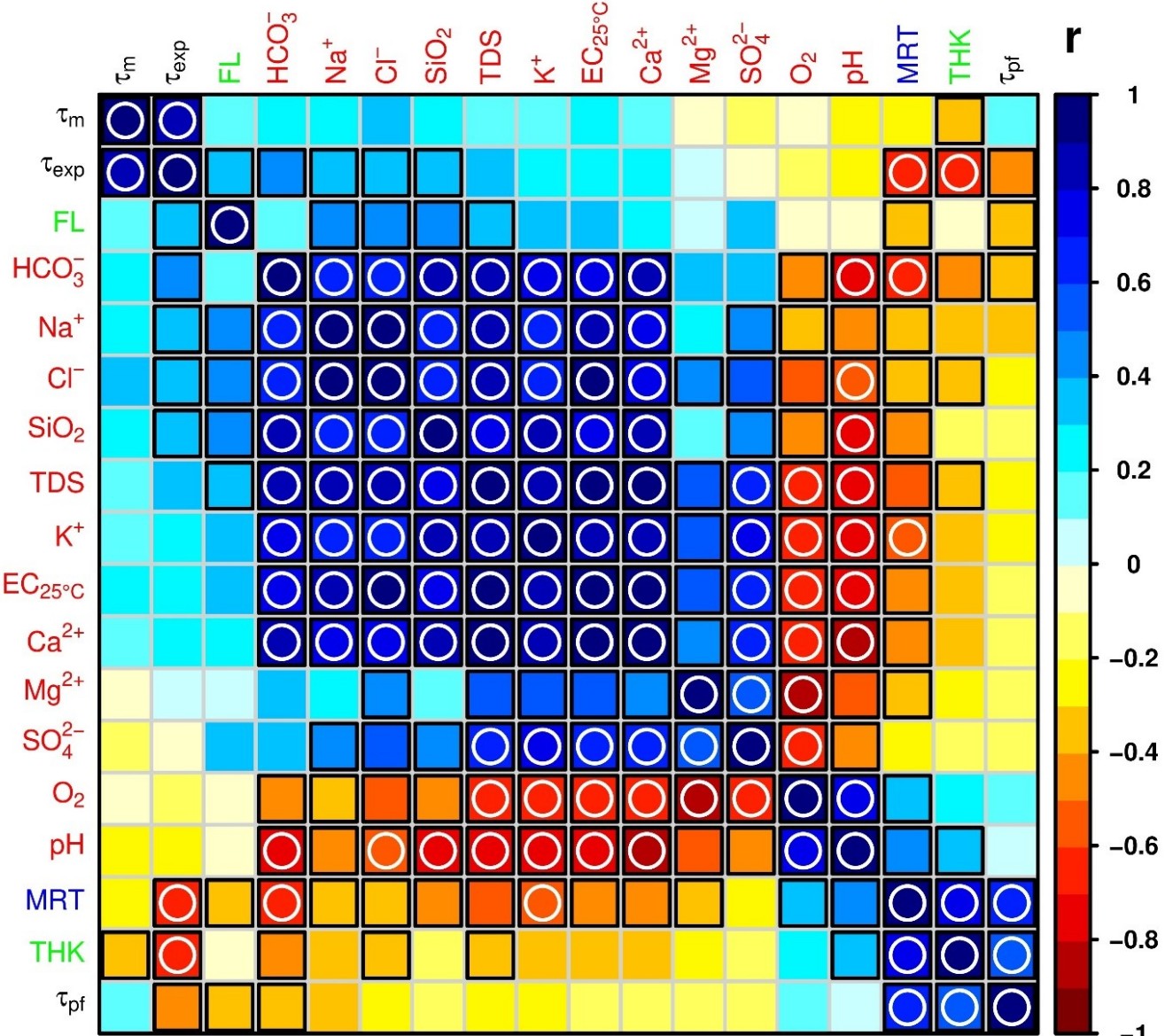

**Figure 12.** Linear correlation (Pearson's r) of tritium derived transit times ($\tau_m$, $\tau_{exp}$ and $\tau_{pf}$ median values of the ensembles of EPM solutions; black labels) with the other springs characteristics, i.e. the hydrochemical parameters average values (red labels), the estimated mean groundwater flow lengths (FL) and the mean Luxembourg Sandstone thickness (THK) of the recharge areas (green labels), and the MRT median values characterizing the response time of spring discharge to effective precipitation input (blue label). A black square denotes a significant correlation (p-value < 0.05), while a white circle highlights a very significant one (p-value < 0.001).

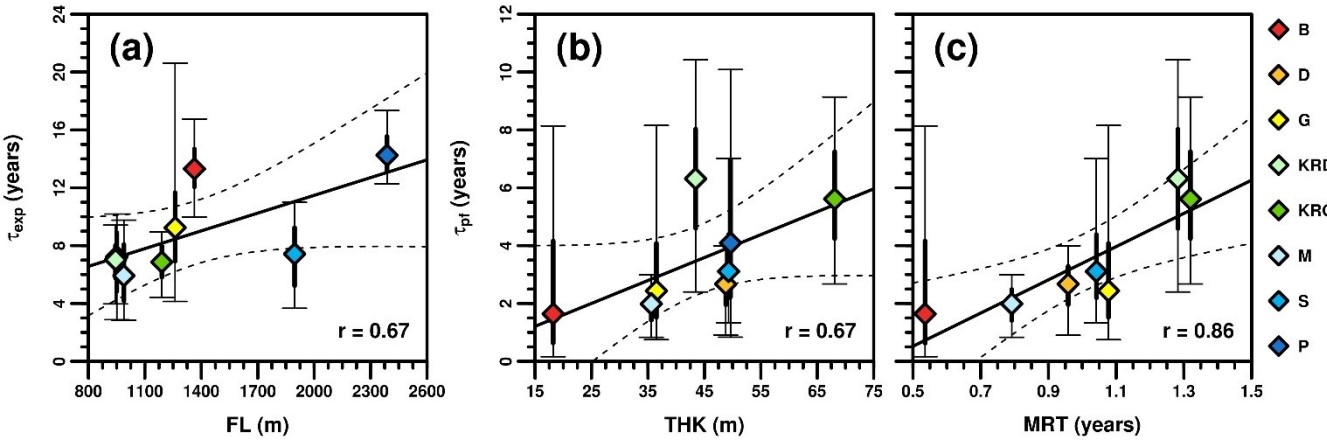

**Figure 13.** Scatter plots relating at the recharge area scale (a) $\tau_{exp}$ median values of the EPM ensembles to the groundwater flow length (FL), $\tau_{pf}$ median values of the EPM ensembles to (b) the mean Luxembourg Sandstone thickness (THK) and (c) the average of the MRT median values characterizing the response time of spring discharge to effective precipitation input. The dashed lines represent the 95% confidence intervals of the regression thick lines. The vertical thick and bounded thin lines represent the interquartile and 5-95th percentile ranges of $\tau_{exp}$ or $\tau_{pf}$ values distributions, respectively.

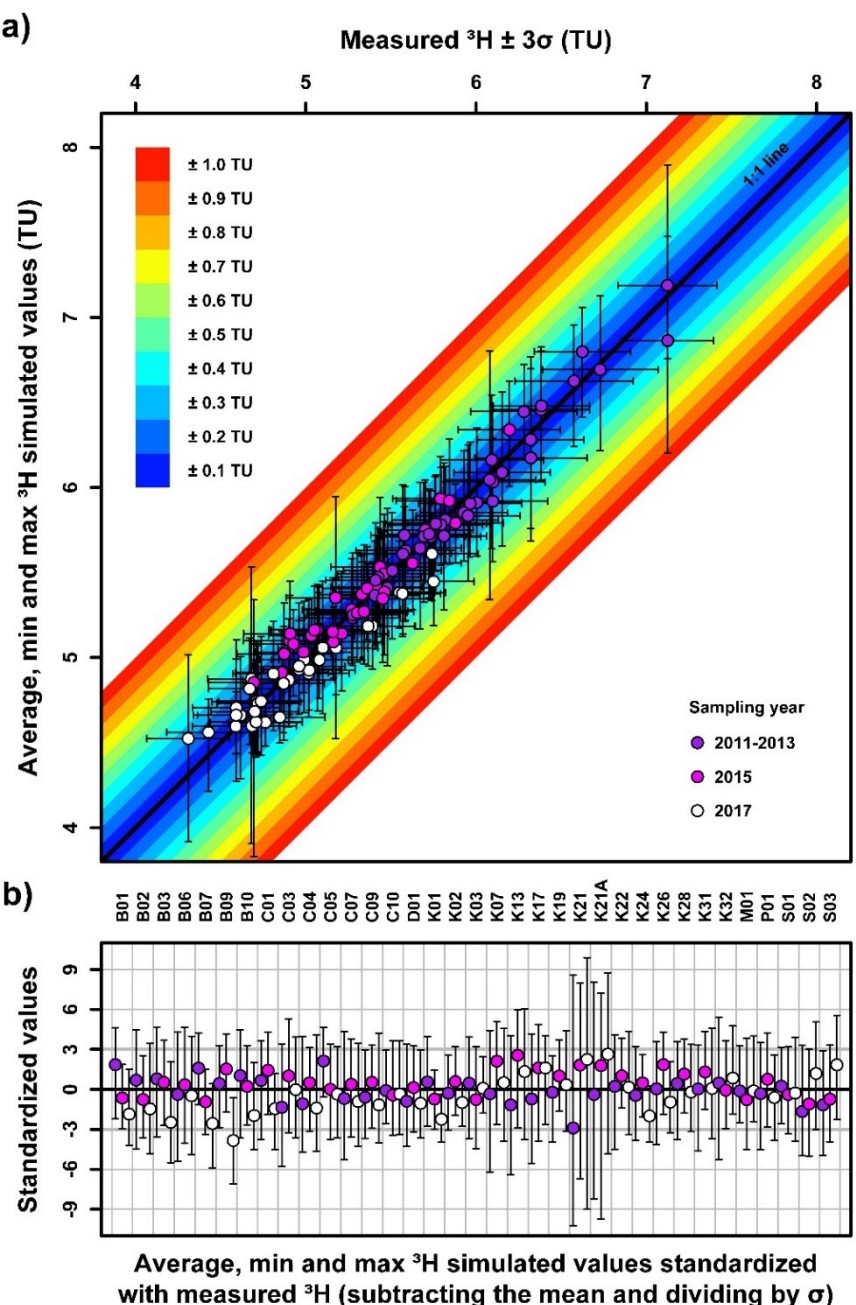

**Figure 14.** Comparison between observed and EPM simulated concentrations in tritium for the 35 investigated springs of Luxembourg City (N samples = 105) as a) a scatter plot and b) standardized values. The root-mean-square error between the observed and average simulated concentrations (colored circles) is 0.11 TU. The min-max range of the simulated concentrations (vertical bounded thin lines) and the $\pm 3\sigma$ range of the measured concentrations (horizontal bounded thin lines) average 0.69 and 0.54 TU, respectively.

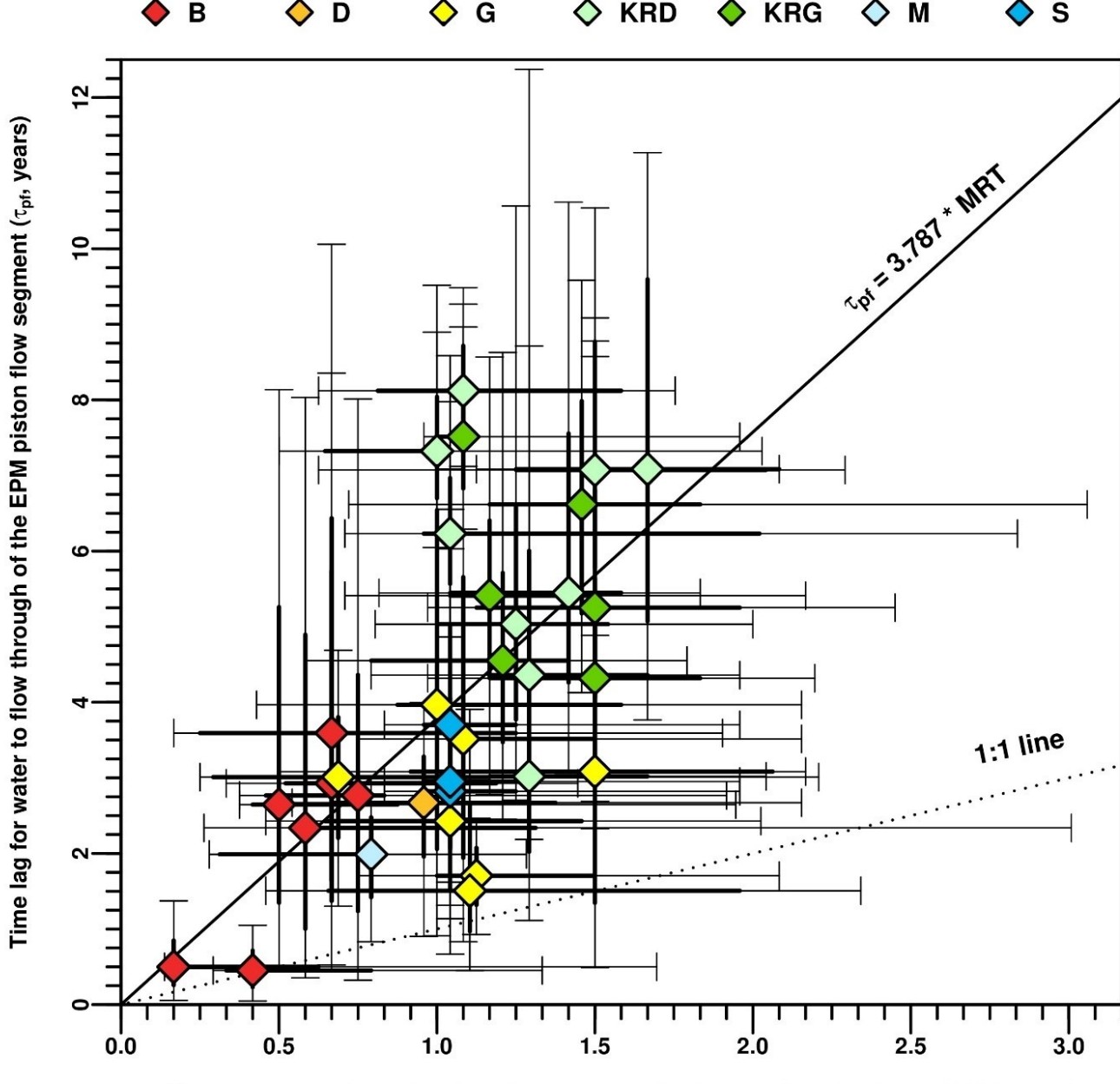

1230

**Figure 15.** Scatter plot relating for each spring $\tau_{pf}$ values of the ensembles of most likely EPM model solutions to MRT values of the ensembles resulting from the effective precipitation – spring discharge cross-correlation analyses. The diamond shapes indicate the medians of the distributions, and their fill colors allow to discriminate the different groups of springs. The thick lines and bounded thin lines represent the interquartile and 5-95[th] percentile ranges of the distributions, respectively.