# Peer review of "Short high-accuracy tritium data time series for assessing groundwater mean transit times in the vadose and saturated zones of the Luxembourg Sandstone aquifer"

_Hydrology and Earth System Sciences, 2023_

## Author Comment (AC1)

**Reply (in blue) to « RC1: 'Comment on hess-2023-152', Anonymous Referee #1, 24 Jul 2023 (https://doi.org/10.5194/hess-2023-152-RC1) » (in italic)**

*This works provides an analysis of tritium observations to estimate groundwater mean transit times in the Luxembourg Sandstone aquifer. The analysis benefits from the interpretation of a large tritium observation dataset that is distributed in both space and time. The authors apply an uncertainty analysis framework to quantify some of the salient sources of uncertainty in estimating mean groundwater transit times from the tritium observations. A powerful component of this work is the assimilation of spring discharge response times and major ion solute chemistry to regularize the tritium analysis. The primary conclusion from the analysis is that groundwater mean transit times can successfully be interpreted at this field study site, despite numerous sources of uncertainty and short observation timeseries. While the manuscript is easy to follow, it often lacks clarity and work is needed to update the language.*

We thank Referee #1 for reviewing our manuscript and providing several constructive suggestions. Despite overall positive feedback, Referee #1 did also express some concerns about our work. We believe that her/his main concerns are partly based on misunderstandings. In this document, we provide our detailed responses to Referee #1's comments and also mention how we plan to address them in a revised version of this manuscript.

*My main concern is that this work does not provide novel nor significant insights into the interpretation of tritium observations nor hydrologic processes at the investigated site. Rather, it is generally an application of established methods to estimate mean groundwater transit times. There is little interpretation into what the inferred mean transit times add to our understanding of this aquifer.*

We are sorry that Referee #1 did not perceive the significant/novel aspects associated with our work.

As the reviewer rightly said, and our article does not contradict this, our work applies existing methods for estimating mean groundwater transit times using tritium. However, our study focuses on investigating new sites, whereas recent studies using tritium for transit time assessments in the Northern Hemisphere have essentially focused on sites that already have historical tritium records and thus benefit from longer time series (see lines 77-82 of our manuscript). This is one of the novel aspects of our work. We show how to face current limitations of tritium dating in the northern hemisphere for characterising new groundwater sites. This is in our opinion valuable for scientists that may like to assess transit times of young groundwater bodies which lack historical tritium data in this part of the world.

While Referee #1 claims the opposite, we also believe that our study provides key learnings on hydrological processes in the Luxembourg Sandstone aquifer. Our assessment of the mean transit times of the springs of Luxembourg city and their relationship with both the vadose zone and the saturated zone of the Luxembourg Sandstone aquifer, as significantly supported by several other types of data, are in our opinion an undeniable asset for better monitoring, regulating, and managing the transport of contaminants in this key drinking water resource.

*Furthermore, it is not clear whether the uncertainty analysis applied in this work results in a substantial advantage over the already established tritium interpretation techniques, many of which also propagate uncertainties.*

As no reference is given, we were unfortunately unable to target the "already established tritium interpretation techniques propagating uncertainties" that Referee #1 is referring to here. To our knowledge, tritium dating studies using lumped parameter models typically follow a basic best-fit calibration approach that does not incorporate tritium analytical uncertainties (which is not appropriate in the context of our study, as argued in the manuscript section 3.1.4, lines 233-243).

*Major Points:*

*While this work propagates the analytic tritium errors in the uncertainty analysis, I feel that it misses the key point that mean groundwater transit times can be non-unique because tritium is insensitive to the long residence time groundwater fractions (the tails of the transit time distribution). This non-uniqueness poses a major challenge in estimating mean transit times with tritium alone and is potentially the larger source of uncertainty compared to analytical errors. I think that the introduction can be revised to better account for the multiple sources of uncertainty in tritium mean transit times and clearly state in what circumstances having a tritium timeseries is most likely to reduce this uncertainty and in what situations it cannot.*

Referee #1 is right; tritium is not sensitive to long residence time groundwater fractions. This information is implicit in our manuscript when we indicate in the introduction the range of mean transit times to which tritium provides access (i.e., the sentence lines 50-52), but this is obviously not clear enough in this version of our manuscript. We propose to rephrase it in the revised version of our article. In this age range (i.e., up to 100-200 years maximum), we nonetheless agree that there is currently a problem of non-unicity of the tritium dating results. But this information is also already mentioned in our introduction (lines 71-80) as well as where/why this ambiguity happens (i.e., in the Northern Hemisphere because the remnant bomb tritium activities have not yet decayed to below those of current rainfall) and how having a tritium time series can help facing this issue (i.e., the longer the time series, the better).

*Estimating groundwater mean transit times with tritium and the convolution integral is not new. Thus, I do not entirely agree with the claim that this work is pioneering interpretation of sparse tritium measurements in the northern hemisphere.*

We assume that Referee #1 is referring here to the final sentence of the introduction (i.e., "By testing the potential of rather short but high-accuracy recent tritium data time series, our work has pioneering character for the dating of young groundwater bodies in Central Europe."; lines 95-96). Whether Referee #1 comment highlights the inappropriate use of the term "pioneering" or a lack of precision (e.g., not specifying that our study concerns only new sites characterisation), note that we will rephrase this sentence in the revised version of the manuscript.

*I suggest revising the manuscript to focus more on the new hydrologic processes we learn from the tritium analysis, rather than inferring mean transit times from tritium. Alternatively, if the more accurate tritium measurements are proving to be essential to estimate mean travel times compared to previous studies, I think this needs to be explicitly shown in this work. While the mean travel time estimates in the Luxembourg aquifer can have value, I feel any innovation in this work is not properly highlighted.*

We do not agree with Referee #1's suggestions. We indeed believe that explaining our rationale to assess mean groundwater transit times of new 'undated' sites using tritium and lumped parameter models is just as important as deciphering transit times in terms of hydrological processes. Besides, these two aspects are inextricable in the context of our study. This is why our discussion section deliberately develops the two aspects in a mixed and balanced way. Moreover, as indicated in our introduction (lines 86-88), the current advantage of using high-precision tritium analyses for obtaining more accurate water dating results has already been demonstrated in previous studies and we thus do not think it is necessary to specifically document it in our study. Additional references on this subject could nevertheless be added to the revised version of the manuscript, for instance Stewart et al. (2010).

*I have concerns on the authors approach to constrain the parameters during the Monte Carlo uncertainty analysis. For instance, the mean transit time parameter is forced to be less than 35 years during the Monte Carlo sampling. Yet, the evidence and rationale to make this assumption is not strong and needs further discussion. Falsifying old transit times due to the high storage volumes is one line of evidence; however, I feel the highly simplified interpretation of storage volumes in this work is not a particularly strong argument and further analysis or comparison to alternative environmental tracers is needed. Furthermore, if it is evident*

*that mean transit times cannot be older than 35 years, it is unclear why the first Monte Carlo was performed. Generally, our prior bounds should encompass our best estimate of the uncertainty ranges for the given parameter. If the reader is to trust the uncertainty analysis, I suggest that the implications of these strong assumptions on the plausible mean transit times need to be further explored and presented.*

As done in several other tritium dating studies (e.g., Morgenstern et al., 2010; Stewart et al., 2010, 2012; Gallart, et al., 2016; Gusyev et al., 2016; Stewart and Morgenstern, 2016), we considered important to explore in our study the full range of mean residence times to which tritium provides access (i.e., 0-200 years). Indeed, we believe that this first broad exploration of the parameter space is key to illustrate comprehensively the current problem of non-uniqueness of tritium dating results in the Northern Hemisphere.

Concerning the justification for restricting the range of residence times in a second stage, we maintain that in our case the rejection of the oldest subpopulation of solutions (i.e., whose mean value varies between 64 and 98 years depending on the springs) due to the excessively high storage volumes that this would imply is a sufficiently solid decision criterion. We agree that Referee #1's concerns would have been justified if a connection with a deeper and more extended aquifer layer had been probable, but this is not the case here. In the context of our study, the Luxembourg Sandstone is isolated from such a connection (because cut by several valleys and perched on an impermeable underlying layer; cf. section 2.2 and Fig. 2). Note that this explanation will complement our revised manuscript.

Referee #1 also states that further analysis or comparison to alternative environmental tracers is needed, but we think this is exactly what we already propose in our work as we use the hydrogeological information at hand, the spring water physico-chemistry, as well as the spring discharge reaction time to effective precipitation inputs to corroborate our final modelling results.

*It is unclear whether the mass balance is closed using the spring discharge measurements alone, or if river discharge or regional groundwater flow are additional fluxes that need to be considered.*

Referee #1 refers here to the water balance calculations mentioned in lines 146-148 of the manuscript, which have been carried out for the different groups of springs used by the City of Luxembourg. As a reminder, these calculations have not been done in this study but were carried out by consultancy firms for the Luxembourg government as part of the official delimitation of the catchment areas of groundwater drinking water resources. To the best of our knowledge, only mean spring flows have been considered for these calculations. It is worth reminding that we also highlighted a potential problem with the recharge area of Group B springs, which seems too small considering our tritium dating results (see lines 544-551).

*In general, there needs to be more explanation of the soil model that calculates the ET fluxes and how the uncertainties in the effective recharge rates impact the mean transit times. This uncertainty can potentially be more important than the tritium analytical error. While it can be argued this point is beyond the scope of this manuscript, I think this work needs to better acknowledge the many sources of uncertainty when quantifying mean transit times with tritium observations.*

See our response to Referee #1 specific comment regarding the soil model parameterization (i.e., L218).

*My opinion is that major revisions are required to address the comments above and to highlight the novelty in this work.*

We hope that the answers we provide in this document will help to dispel Referee #1 concerns.

*Specific Comments:*

*L19: I suggest that the abstract needs to explicitly state what the identified mean transit times are.*

We will modify the abstract according to Referee #1 comment.

*29: Duplicated references.*

This is not duplicated references, this is two different ones.

*L34: The use of 'by far' and 'obviously' are too strong of descriptions and are not needed. These are examples of the many places where the clarity of the manuscript can be improved.*

In the revised version of the manuscript, 'by far' and 'obviously' will be removed.

*L67: I think at this point the Introduction needs to mention the power of 3H/3He dating, which has largely super-seeded using 3H alone. Using 3H alone represents a major limitation of this study in my opinion.*

We do not agree with Referee #1. As explained a few lines before in the introduction (lines 53-59), the use of the $^3H/^\beta He$ dating technique faces issues specific to gaseous tracers which we wanted to avoid in our study. First, water sampling in spring locations is particularly thorny due to the risk of contamination with the surrounding air. But above all, as $^3He$ generated in the vadose zone is effectively lost from the water phase, the $^3H/^\beta He$ clock does not start until water is below the water table and completely isolated from a gas phase. In our study, we didn't want the vadose zone to be neglected.

*L208: The information "initiated in 1958 by the IAEA (International Atomic Energy Agency) and the WMO (World Meteorological Organization)" is not needed.*

This information will be removed in the revised manuscript.

*L218: Why was a water-holding capacity of 100 mm assumed? In general, there needs to much more information on the soil model.*

This value is an overall mean value derived from laboratory measurements carried out on samples taken from several soil pits and is usually used for soils overlying the Luxembourg Sandstones (Hissler et al., 2015; Hissler and Gourdol, 2015). This information will complement the revised version of the manuscript. In addition, as shown in the figure hereafter, it should be noted that using a soil maximum water-holding capacity varying between 50 and 150 mm does not have a strong impact on the tritium recharge input.

[Figure]

*L225: Equation 3 for the tritium input weighting does not seem correct. Will Peff just not cancel out because it is in the numerator and denominator unchanged?*

Referee #1 misunderstood Equation 3.

If the equation had been:

$$C_{in\ i} = \frac{\sum_{j=i-11}^{i} C_j \times \sum_{j=i-11}^{i} C_j Peff_j}{\sum_{j=i-11}^{i} Peff_j}$$ , then yes there would indeed have been cancellation.

But the equation is:

$$C_{in\ i} = \frac{\sum_{j=i-11}^{i} C_j Peff_j}{\sum_{j=i-11}^{i} Peff_j}$$ , which is correct.

For instance, for i = 12:

$$C_{in\ 12} = \frac{\sum_{j=1}^{12} C_j\ Peff_j}{\sum_{j=1}^{12} Peff_j} = \frac{C_1 Peff_1 + C_2 Peff_2 + C_3 Peff_3 + \cdots + C_{10} Peff_{10} + C_{11} Peff_{11} + C_{12} Peff_{12}}{Peff_1 + Peff_2 + Peff_3 + \cdots + Peff_{10} + Peff_{11} + Peff_{12}}$$

*L241: It states that the input signal uncertainties are propagated by randomly sampling from a normal distribution; yet there is no information on what the distribution parameters are.*

Means and standard errors values of the recharge input signal used to generate the replicated time-series are shown in Fig. 5b,d. These values will also be supplied in a dataset and archived in an online repository, whose DOI will be included in the final published version of the paper.

*L255: It is not explained why the comparison to hydraulic response times and major ions is done in first place. Adding some rationale for why this analysis was performed into the introduction seems necessary.*

We thank Referee #1 for this advice. We have chosen to include this information in the "Material and methods" section so as to not overload the introduction. But if required, the reason for using the spring chemistry, as well their hydraulic response time to the precipitation inputs, will be additionally mentioned in the introduction section of the revised version of the manuscript.

*L345: It is not clear how the measured tritium suggests the presence of bomb-peak water. For instance, it seems that the measured values could be obtained through recent recharge and radioactive decay.*

We agree with Referee #1, it is theoretically possible for some springs considering the modelling results presented in the Supplementary file. Note that in the revised version of the manuscript, the last part of the sentence (i.e, "emphasizing the presence of bomb tritium in the Luxembourg Sandstone aquifer"; line 346) will be removed.

*L357: I think there needs to be much more discussion on how these samples with old mean transit times can be confidently rejected. I am not convinced of this assumption given the terse description provided.*

See our response to Referee #1 general comment.

*L366: Why were only a third of the samples retained? The seemingly subjective 'throwing away' of samples concerns me and seems to deteriorate the uncertainty quantification significantly. The assumptions and rationale need to be explained better. L369: Same comment as above. Is keeping only the dominant population actually providing a robust measure of the uncertainty?*

Referee #1 is concerned here about our approach to use an additional 2D kernel NSE weighted density filter to obtain a more coherent ensemble of solutions from the raw modelling results obtained from the second EPM parameter space screening and asks for clarification.

As shown in the Supplementary material (Fig. S1-S35) and mentioned in the manuscript (lines 363-366), the 5000 behavioural solutions resulting solely from the NSE threshold of 0.5 were not distributed around an unambiguous parameter distribution, but spread into several subpopulations - most of the time overlapping, and thus leading to an ensemble with a complex multimodal character. We wanted to isolate the most probable unimodal distribution of parameters to tend toward, what we called in line 366, "a more coherent ensemble of solutions". It is to face this issue that we opted for a 2D kernel NSE weighted density filter. In practice, its use allowed the multimodal distribution to retain the (almost) unimodal set of solutions with respect to both EPM parameters with the highest probability (referred to as the "most likely solution ensemble" in the manuscript), and it provided acceptable results for all springs. The decision to keep only a third of the samples was taken, after some trials, as a good compromise on the one hand to reject the least possible solutions and to limit cases requiring rejection of a secondary weaker subpopulation on the other hand. This will be explained in the revised version of the manuscript.

It is true that the use of such an additional filter has a considerable impact on the dating results provided and their associated uncertainties (compared to if we had used all 5000 solutions), but, as mentioned in lines 502-503 and supported with Fig. 14, the simulated spring tritium concentrations resulting from theses ensembles are overall consistent with the tritium measurements of spring samples, their downward trends, as well as with their analytical uncertainties. To better document this statement, we propose to add a second graphical panel to Fig.14 that will show the simulated concentrations in tritium standardized with the tritium observations (subtracting the mean and dividing by standard deviation) as in the figure hereafter. Moreover, as mentioned in the discussion section of the manuscript, the distribution of the most probable parameters is supported by several independent data sources (lines 514-521) and we thus believe that our decision is further reinforced.

[Figure]

*L472: This sentence is awkward and is not needed.*

We will follow Referee #1's advice and remove this sentence in the revised version of the manuscript.

*L522: Is this assumption of vadose zone that is the entire thickness of the aquifer reasonable? I think that much more information about the vadose is needed. For instance, are there any wells that can constrain the vadose zone thickness?*

We think that this comment/question relates to a misinterpretation of our statement. We do not assume here that the thickness of the vadose zone is equal to the full Luxembourg Sandstone thickness, but we state that the Luxembourg Sandstone thickness can be considered as a "proxy" of the thickness of the vadose zone (which is different). In statistics, a proxy is a variable that is not in itself directly relevant, but that serves in place of an unobservable or immeasurable variable. For a variable to be a good proxy, it must have a close correlation (either positive or negative, and not necessarily linear) with the variable of interest. The validity of our hypothesis can be supported to some extent with the graph shown hereafter which compares our estimates of the mean Luxembourg Sandstone thickness for each recharge areas from Table 1 (x-axis of the graph hereafter) to the mean vadose zone thicknesses that can be derived from the tritium results (subtracting $H_{exp}$ values indicated in Fig. 8 from the mean Luxembourg Sandstone thickness values in Table 1; y-axis of the graph hereafter).

However, we agree with Referee #1 that it would have been preferable to rely on robust direct information to assess the vadose zone thickness for each recharge area. Unfortunately, as mentioned in lines 148-149, hydrogeological drillings in the Luxembourg Sandstone are too sparse and poorly distributed (one can also see the "Hydrogeological drillings" layer in the Luxembourg platform for governmental geodata and service https://map.geoportail.lu/). For instance, only one borehole is available in the KRD recharge area and none in the KRG, M, P and D ones (excluding shallow exploratory boreholes drilled in the very near vicinity of spring catchment facilities). However, it is worth noting that Farlin et al (2013a) indicate that the saturated and vadose zones are about 10 m and 45 m thick, respectively, at the level of the observation borehole in the KRD sector, which is consistent with the average numbers we assessed in our study (see Fig. 8).

[Figure]

References used in our response to Referee #1 that were not already cited in the manuscript:

Hissler, C., Gourdol, L., Juilleret, J., Marx, S., Leydet, L., and Flammang, F.: Pedotransfer functions for predicting soil hydrological characteristics in Luxembourg: literature review and reliability tests for predicting the soil maximum water-holding capacity (Fonctions de pédotransfert pour la prédiction des caractéristiques hydriques des sols au Luxembourg : analyse bibliographique et premiers tests de fiabilité pour la prédiction de la réserve utilisable maximale des sols), Report drafted on behalf of the Administration des services techniques de l'agriculture (in French), 2015.

Hissler, C., and Gourdol, L.: Assessment of soil maximum water-holding capacity in Luxembourg at national scale: a first estimate based on recent datasets (Évaluation de la réserve utile maximale en eau des sols au Luxembourg à l'échelle nationale : une première estimation basée sur des jeux de données récents), Report drafted on behalf of the Administration de la gestion de l'eau (in French), 2015.

Stewart, M.K., Morgenstern, U. and McDonnell, J.J.: Truncation of stream residence time: how the use of stable isotopes has skewed our concept of streamwater age and origin, Hydrological Processes, 24, 1646-1659, https://doi-org.proxy.bnl.lu/10.1002/hyp.7576, 2010.

Gusyev, M. A., Morgenstern, U., Stewart, M. K., Yamazaki, Y., Kashiwaya, K., Nishihara, T., Kuribayashi, D., Sawano, H., and Iwami, Y.: Application of tritium in precipitation and baseflow in Japan: a case study of groundwater transit times and storage in Hokkaido watersheds, Hydrology and Earth System Sciences, 20, 3043–3058, https://doi.org/10.5194/hess-20-3043-2016, 2016.

---

## Author Comment (AC2)

**Reply (in blue) to « RC2: 'Comment on hess-2023-152', Anonymous Referee #2, 18 Sep 2023 (https://doi.org/10.5194/hess-2023-152-RC2) » (in italic)**

_General remarks_

_The paper deals with a relevant topic. It is well written, straight to the point and interesting for both the specialists and the less experienced audience. For these reasons I recommend minor revisions, that mostly concern the description of the aquifer properties of the Luxembourg Sandstone Formation. As a matter of fact, sedimentary, diagenetic, structural and geomorphological (i.e. diffuse karstification) properties have been too poorly described, sometimes in a misleading way. I suggest to revise deeply section 2, removing the misleading sentences, introducing the most relevant hydrostratigraphic and hydrogeological features and completing/updating the references to the most relevant papers. This will make more convincing the discussion and conclusion sections, permitting to establish reliable comparisons between the modelling results and the heterogeneity and anisotropy of the real world aquifer._

We thank Referee #2 for a thorough review of our research work. We greatly appreciated her/his positive feedback and welcomed all the valuable advice given to improve our manuscript. Please find in this document our responses to Referee #2's comments and how we plan to address them in a revised version of this manuscript.

_Specific remarks_

_Section 2.1_

_The part of this section dedicated to the geological properties of the Luxembourg Sandstone is definitely too poorly informative and insufficient to characterize the heterogeneity and the anisotropy of the aquifer under investigation. It also contains some incorrect statements (see the following remarks). Please update the references about the formation, that are incomplete and in some cases outdated. The most relevant stratigraphic, sedimentological, compositional and diagenetic properties should be mentioned to describe shortly how they control the modes and paths of groundwater flow through the porous/fractured/karstic medium. The superimposed structural pattern of faults and fractures should be also introduced to mention how it contributes to the duality of groundwater circulation (fissured - porous rocks) that You assess. In its present from the description of fractures is almost useless. The presence of widespread karst features, reported by some literature, should be mentioned and commented, also considering the impact of these features on the "dual" circulation system (is it really dual?). In addition, the presence of aquitards within the aquifer group should be introduced before the discussion section. In its present form this section conveys the wrong idea of an almost homogeneous and isotropic sandstone body, with uniform facies/hydrostratigraphic properties through space, that is not the case. It should also be mentioned which members, i.e. which aquifer systems within the group, and at which sites have been sampled and studied. This latter part totally relies on the technical notes by public agencies, that do not permit to figure out the geometry of the aquifers composing the group, of the compartments within them and of their recharge areas. As a matter of fact, the identification of the groundwater bodies that You are studying is not sufficiently clear to the reader and reliable. As You state in Your discussion section, the integration of Your modeling results with the knowledge on aquifer geometry and physical behavior would lead to "more accurately represent the multi-scale complexity of the Luxembourg Sandstone bedrock aquifer". So why don't You start by incorporating a very short synthesis of the most relevant hydrostratigraphic knowledge, in terms of heterogeneity and anisotropy of the rocks and identification of the major groundwater bodies within them in Your paper?_

We thank Referee #2 for these constructive comments on Section 2.1. We agree that some aspects of the Luxembourg Sandstone are not accurately enough depicted or missing in our description, which may lead to

a wrong understanding of the aquifer's context. The section will be modified accordingly in the revised version of the manuscript.

*Line 102: no need to indicate SiO2 and CaC03, just state "quartz" and "calcite"*

In the revised version of the manuscript, "SiO$_2$" and "CaCO$_3$'' will be removed.

*Line 103: really the Luxembourg Sandstone is just a calcite-cemented pure quartzarenite (see for instance Berners, 1983)? is the average bulk chemical composition relevant to describe aquifer heterogeneity? The Luxembourg sandstone is an outstanding example of how diagenesis determined the poro-perm properties under control of the composition of the framework grains and the changes of texture and sedimentary structures (i.e. facies associations).*

We thank Referee #2 for this insightful comment. This will be considered when revising the Luxembourg Sandstone description.

*Line 103: please replace the reference to the unpublished PhD thesis with the reference to Van Den Brill and Swennen (2009), or at least quote also the published paper.*

We thank Referee #2 for this advice. This will be considered in the revised version of the article.

*Line 104: what do You mean by "… crossed by beds of sandy marls"? Are these beds neptunian dykes? Please describe the stratigraphy of the formation properly: there are marly units separating sandstone bedsests, facies and compositional changes (framework grains, cements, matrixes) occur through the sandstone divisions, many bedsets are almost limestones, owing to primary composition and diagenetic replacement, so karst features are widespread in some bodies at some sites (see for instance Meus and Willems, 2021). Do You really would describe this formation as a "uniform" unit? On the contrary it is highly heterogeneous and anisotropic, that implies relevant bearings on Your experiments.*

We thank Referee #2 for this pertinent advice. This will be considered when revising the Luxembourg Sandstone description.

*Line 115: Meus and Willems (2021) is missing in the reference list*

We thank Referee #2 for pointing out this mistake. The full reference will be added to the reference list in the revised version of the manuscript.

*Line 131: considering the regional geology, are You pretty sure that recharge occurs only through the outcrop area of the Formation?*

Correct. The outcrop area of the Luxembourg Sandstone constitutes the main, but not the only, aquifer recharge zone (i.e., additional secondary recharge components exist, e.g., leakages from the overlying Strassen formation, connexions through faults with other geological layers). This will be accounted for when revising the Luxembourg Sandstone description.

*Fig.1 is almost useless to describe aquifer architecture and heterogeneity. Stratigraphic logs introducing the general features of the formation should be added.*

We thank Referee #2 for this advice. A stratigraphic log introducing the general features of the Luxembourg sandstone will replace or complement Figure 1 in the revised version of this article.

*Fig.2 is very difficult to read. The geological attributes are hidden by the elevation map of the formation base (please specify m "above sea level" in the color scale). The legend of the geological features is obscure (alluvial materials? What do You mean? Quaternary? Which formations are involved?). Where do we read these features? Where are represented the fault/fracture systems? Which hydrogeological features are shown? The sampled springs are sparse at different settings. Which units of the Formation have been sampled? All belong*

*to the same sandbar systems? Are some springs located in the limestone (karstified) units? Are there marlstone beds (aquitards) at some locations?*

We thank Referee #2 for raising here readability/accuracy issues with Figure 2. These aspects will be considered in the revised version of the manuscript.

Section 2.2

*Fig.3. This is a very general and unrealistic conceptual picture of the Luxembourg Sandstone Aquifer. It is portrayed as a uniform, homogeneous and isotropic medium, without bedding planes associated to litho-textural variations. Fractures are not drawn (the reader must assume two orthogonal vertical sets everywhere). "Slow infiltration through matrix" is declared in the unsaturated zone, with black lines maybe indicating strange infiltration paths (if I understand the picture) that would never permit the percolating water to reach the saturated zone. Moreover, what is the matrix? In the karstic aquifers, matrix is sometimes intended as the impervious rock with no circulation, that instead occurs through conduits, caves and fractures. In the list of "some numbers" You declare up to 40% porosity, so this would not be a matrix, neither from the lithological/sedimentological point of view (it would be a mudrock) nor from the hydrological point of view. In addition, it looks a little bit strange the use of the conceptual image of a carbonate karstic aquifer, without considering the karstic features of Your specific setting. I strongly suggest to redraw a realistic conceptual model of Your aquifer, with the true stratigraphic, lithotextural, structural and geomorphological features and inserting the plausible location of the clusters of springs You sampled. Please note that I am not asking for a more detailed or accurate picture, I just would like to see a very simple and general model showing the most relevant features of Your aquifer group.*

As proposed by Referee #2, we will strive to draw a simpler/general but more realistic hydrogeological conceptual diagram in the revised version of the manuscript.

Section 3.1

*Lines 176-177: this statement should have been supported by the description of the conceptual model of the Luxembourg Sandstone in Chapter 2, that is unfortunately largely insufficient. Moreover, this assumption should be site-specific in such a large aquifer group as the one You are dealing with.*

This statement will be supported by the revised Section 2.

*Line 190: Fig.2 does not explain the hydrogeological setting of the 32 sampled springs. Do they share the same recharge area, geological and hydrological conditions? Do the same approach apply to all the sampled springs? You rightly mention the hydrochemical and hydrogeological and exploitation variability among them, that should be better described and accounted for in Your approach.*

As mentioned previously in our response to Referee #2's comments, Figure 2 will be modified to be more readable. The revised figure will also indicate more precisely which sampled spring belong to which recharge area. Referee #2 asks also here if "the same approach apply to all the sampled springs?", but unfortunately, we have not been able to understand what this refers to.

*Lines 200 and following: which data on average thickness and presence/absence of surface soils did You use? Which data for evapotranspiration?*

We assume that Referee #2 is referring here to "lines 220 and following" and asks for clarification about the approach used to assess effective precipitation. As mentioned in the manuscript (see lines 218-223 & 300-303), we followed the Thornthwaite method (1948). Perhaps our explanation is not clear enough for the reader and we will rephrase it in the revised version of our article.

The Thornthwaite method is essentially a water balance of the rootzone performing monthly book-keeping of precipitation, evapotranspiration and soil moisture. Deep infiltration below the root zone (i.e., effective

precipitation) occurs only when field capacity is exceeded (i.e, the maximum water-holding capacity). The Thornthwaite method (1948) comes with an empirical potential evapotranspiration formula using only monthly mean air temperature as input. Although this approach is empirical and could be considered outdated, the Thornthwaite method is still widely accepted and used in several disciplines, especially in hydrogeology for estimating aquifer recharge (e.g., Lanini et al., 2016; Mammoliti et al., 2021).

As already mentioned in our response letter to Referee #1's comments, the maximum water-holding capacity set to 100 mm in our study is an overall mean value derived from laboratory measurements carried out on samples taken from several soil pits and is usually used for soils overlying the Luxembourg Sandstones (Hissler et al., 2015; Hissler and Gourdol, 2015). As asked by Referee #2, it is worth noting that soil pits indicate soils 40 to 100 cm thick (average 83 cm).

*Lines 245-250: an effort to estimate the aquifers volumes in order to obtain some independent numbers to evaluate the estimates of groundwater volumes would make this study a little more linked to the real world.*

We agree with Referee #2 that it would have been nice to rely on direct data that would have made it possible to assess the volumes of groundwater stored in the aquifer in a different independent way. For instance, measurements of the water table level in boreholes could have made it possible to estimate the volumes of water stored in the saturated zone of the aquifer. Unfortunately, as mentioned in lines 148-149, hydrogeological drillings in the Luxembourg Sandstone are too sparse and poorly distributed (one can also see the "Hydrogeological drillings" layer in the Luxembourg platform for governmental geodata and service https://map.geoportail.lu/). For instance, only one borehole is in the KRD recharge area and none in the KRG, M, P and D ones (excluding shallow exploratory boreholes drilled in the very near vicinity of spring catchment facilities). However, it is worth noting that Farlin et al (2013a) indicate a saturated thickness of about 10 m at the level of the observation borehole in the KRD sector, which is consistent with the average numbers we assessed in our study (see Fig. 8).

*Section 3.2*

*Lines 289 – 290: here You refer to the current use of effective infiltration in karst aquifers after literature, but since this statement You did not consider the Luxembourg sandstone a karst aquifer. This issue must be addressed properly in section 2 where You should definitely characterize the aquifer group under investigation as karstified or not.*

We thank Referee #2 for this comment. The karstification degree of the Luxembourg Sandstone (which is overall relatively poor despite the presence of karstic features) will be described in the revised Section 2.

*Lines 296-297: this assumption, in my opinion, is unrealistic and makes poorly reliable the use of effective infiltration.*

Referee #2 disagrees here with our decision to consider the effective precipitation homogeneous for the entire study area, but without explaining why. As argued, the spatial extension of our study area is relatively restricted. This allows us to assume that the spatial variability of the precipitation input signal is rather small (especially since precipitation is effective mainly in winter, a period during which the spatial variability of precipitation fields is particularly low in comparison to convective summer rainfall event; one can also see the work of Pfister et al. (2017) describing the spatial variability of the precipitation as rather homogeneous over a region otherwise even wider containing our study area). In addition, recharge area mean elevations of the different groups of springs are very close to each other (see Table 1), which makes it possible to consider similar temperatures from one area to another. Perhaps Referee #2 does not agree with our hypothesis because thinking that the soil type/cover variability would induce too much heterogeneity in the effective precipitation input signal. If so, here again we think that our assumption is reasonable. As element of justification, the figure hereafter documents the yearly moving effective precipitation computed for a maximum water-holding capacity varying from 50 to 150 mm (which allows to some extent the soil variability

to be mimicked). It is true that the maximum water-holding capacity value impacts the effective precipitation amount (the smaller the maximum water-holding capacity, the higher the effective precipitation; see the upper panel of the figure), but this does not impact the temporal variability of the signal (see the lower panel of the figure) which is finally the most important for the cross-correlation analysis performed in our study.

[Figure]

*Line 311: I suggest to make explicit the abbreviations at least at their first appearance (MRT)*

Note that the MRT abbreviation is already introduced in the manuscript line 276.

*Lines 323-324: might You consider the opportunity of inserting the Piper plots to better visualize and characterize the eventual variability of hydrochemical facies? At line 329 You state that the "… the spatial variability of the hydrochemistry … is stable over time… ", implicitly admitting that this variability do exist and might be clarified by these very simple and traditional plots.*

We have already tested the potential added value of a Piper diagram (which allows to distinguish the major types of water based on major ions relative composition), but we concluded that this type of representation would not be the best option for documenting the hydrochemical variability in our study. Indeed, the springs of Luxembourg City, although exhibiting an inter-spring hydrochemical variability, are all characterized with the same $CaHCO_3$ facies and are therefore represented very close to each other in a Piper plot (see figure hereafter).

[Figure]

Piper plot of the spring mean physicochemical characteristics provided in Table S3

_Section 4_

_Lines 348 – 350: the recharge areas of the spring groups are not sufficiently described and commented by the previous sections and by Fig.2 that are based on the 1939 geological maps and on local RGD technical notes, that make it very difficult to obtain a reliable idea of the groundwater bodies and hydrogeological basins involved by the study._

See our responses to Referee #2's comments about Section 2, Figures 2 and 3, and Line 190.

_Section 5_

_Through all the discussion section, several different properties of the real aquifer are mentioned for comparisons with the modelling results and the assessment of uncertainty (real sandstone thickness, vadose/saturated zone ratio, modes and times of transit through the vadose and saturated zones of the dual porous/fractured aquifer without considering the karst features, presence of local or widespread confining layers and internal compartments in the aquifer group, areas and lengths of the recharge regions and paths, hydrochemical properties and so on). In most cases You claim that these comparisons support the results, but most of these properties have been introduced and considered in a very rough and generic way, sometimes not truly coincident with the real world. In addition, You formerly used many of the same properties to take or to validate decisions, so introducing some circularity in Your line of reasoning. As a matter of fact, this generic use of a poorly presented knowledge on the aquifer group does not really support the results, on the contrary highlighting to the reader the distance between knowledge on the real-world aquifer heterogeneity and anisotropy and the presented modeling results. An example is given by the last part of the section (from line 552 to the end) where the obvious anisotropy of this kind of aquifers, that is portrayed by hundreds of papers of the current hydrostratigraphic literature, is introduced and discussed very roughly. Over these lines the Authors look to discover this property (that is shared by all the bedded aquifers) at this point of the paper (see also lines 590-592 in the Conclusions), arguing that this physical property might be better understood, so it should be studied in order to set up an integrated model (hydrostratigraphic, hydrogeological,_

*hydrochemical, let's say in 4D), that incorporates the tritium based LPM approach nicely proposed in this paper. So, why the Authors did not use all the existing literature on the Luxembourg sandstone aquifer group to make tight comparisons between the real world and their modelling results? How nice would have been to show that the well-known anisotropy of the bedded aquifer is mirrored by the computed anisotropy of water velocity through the vadose and saturated zones?*

We thank Referee #2 for these constructive comments. Echoing the revision of Section 2 (which will notably introduce the anisotropy of the bedded aquifers to which the Luxembourg Sandstone belongs), it is worth noting that several parts of the Section 5 Discussion will be rewritten in the revised version of the manuscript.

*Lines 475 – 476: I agree that presuming stationarity for groundwater is less critical than for stream water, but this does not mean that steady-state might be assumed safely for a heterogeneous, mixed karstic/fractured/porous aquifer group like the Luxembourg sandstone.*

As Referee #2 concedes, presuming stationarity for groundwater is in general less critical than for stream water. Including a statement mentioning it is thus reasonable in our opinion, and we would therefore prefer to keep our sentence as is.

References used in our response to Referee #2 that were not already cited in the manuscript:

Hissler, C., Gourdol, L., Juilleret, J., Marx, S., Leydet, L., and Flammang, F. : Pedotransfer functions for predicting soil hydrological characteristics in Luxembourg: literature review and reliability tests for predicting the soil maximum water-holding capacity (Fonctions de pédotransfert pour la prédiction des caractéristiques hydriques des sols au Luxembourg : analyse bibliographique et premiers tests de fiabilité pour la prédiction de la réserve utilisable maximale des sols), Report drafted on behalf of the Administration des services techniques de l'agriculture (in French), 2015.

Hissler, C. and Gourdol, L.: Assessment of soil maximum water-holding capacity in Luxembourg at national scale: a first estimate based on recent datasets (Évaluation de la réserve utile maximale en eau des sols au Luxembourg à l'échelle nationale : une première estimation basée sur des jeux de données récents), Report drafted on behalf of the Administration de la gestion de l'eau (in French), 2015.

Lanini, S., Caballero, Y., Seguin, J.J., and Maréchal, J.C.: ESPERE – A Multiple-Method Microsoft Excel Application for Estimating Aquifer Recharge, Groundwater, 54, 155-156, doi.org/10.1111/gwat.12390, 2016.

Mammoliti, E., Fronzi, D., Mancini, A., Valigi, D., and Tazioli, A.: WaterbalANce, a WebApp for Thornthwaite – Mather Water Balance Computation: Comparison of Applications in Two European Watersheds, Hydrology, 8, 34, doi.org/10.3390/hydrology8010034, 2021.

Meus, P. and Willems, L.: Tracer tests to infer the drainage of the multiple porosity aquifer of Luxembourg Sandstone (Grand-Duchy of Luxembourg): implications for drinking water protection, Hydrogeology Journal, 29, 461–480, doi.org /10.1007/s10040-020-02274-z, 2021.

Pfister, L., Martínez-Carreras, N., Hissler, C., Klaus, J., Carrer, G.E., Stewart, M.K., and McDonnell, J.J.: Bedrock geology controls on catchment storage, mixing, and release: A comparative analysis of 16 nested catchments, Hydrological Processes,31, 1828–1845. doi.org/10.1002/hyp.11134, 2017.

---

## Author Response (AR2)

We are pleased to send you the final version of our manuscript. We have taken into account the last comments and suggestions of Referee #1 and Referee #2. Our responses (**in blue**) to their comments (***in italic***) are detailed below. Note that the Code and data availability section has also been modified and includes now the DOI of the repository archiving the R scripts and data used for this study.

**Response to Anonymous Referee #1**

*The comments from the previous review have largely been addressed in the responses and manuscript. Concerns regarding novelty of this study also have been adequately articulated and discussed. The manuscript generally reads well and contains detailed descriptions of theory and methods, making this work suitable for a wide audience. I still have minor comments, largely related to clarifications and writing clarity. Details are below.*

*Comments:*

*- It would be good to know the ET values that were used to correct the precipitation signal, and how these ET compare to other studies in the region (likely in SI). It seems like using effective precipitation aided the interpretation of this study, but it is hard to evaluate how realistic the Thornthwaite's ET values are.*

Thornthwaite's potential evapotranspiration values used to correct the precipitation signal are accessible following the DOI indicated in the Code and data availability section. Note that Pfister et al. (2017) compared in a region encompassing our study area monthly potential evapotranspiration data obtained via both the FAO-reference  Penman–Monteith and Thornthwaite formulas. Their results show that the difference is less than ±5%. This information was added to the manuscript at the end of the second paragraph of section 3.1.3.

*- It would be helpful to discuss some of the issues with tritium (and in general only using a single tracer). For instance, aggregation errors in tritium can bias estimated mean travel times young (Bethke & Johnson, 2008). Without additional tracers, the full transit time distribution remains unknown, and the mean can be biased.*

We followed this advice. New elements have been added at the end of the discussion section in this respect.

*Line Comments:*

*Abstract: The content looks good; however, there remains grammatical and language errors. I suggest further review on the language.*

*L9: manifold 'of' ...*

*L25: delete 'obtained'*

*L28: delete 'particularly well'*

*L29: improve to improving*

*L30: delete 'on a larger scale'*

*L38: I had to read this long sentence multiple times to understand. I suggest breaking into multiple smaller sentences.*

*L65: Suggest replacing 'thorny' with challenging or some other word.*

*L65: '... risk of sample contamination...'*

*L66: Consider removing 'rapid exchange with the atmosphere after emergence' as it is basically stated just prior.*

*L71: Suggest deleting 'first to carry out'*

*L91: Suggest revising this sentence for clarity: "...an important aspect that must be considered is the tritium measurement accuracy..."*

We agree with all "Line comments" above and have modified the manuscript as suggested.

*L167: "it is assumed not to dominate water transit times in the aquifer at large scale". Why is this the case? Is this in contrast to studies that show fast-flowing components dominate the flux-weighted transit time distribution in streams (e.g. Berghuijs & Kirchner, 2017)?*

We agree that a reference to support this hypothesis is missing. We added at the end of the sentence the reference to Farlin et al. (2013a), who concluded on a rather small relevance of the fast flow component in the Luxembourg Sandstone aquifer (based on a time series analysis of spring water chemistry and stable isotopes).

*L195: Suggest dropping 'consultancy firms' and just citing the reports as normal.*

We followed this advice.

*L410: To clarify, are the plotted spring tritium decay-corrected?*

The tritium data are decay corrected to the date of sampling and then plotted as at the date of sampling, as it is standardly done.

*Could the downward trend be explained by the sampling month, given there is large variation in the intra-annual tritium input based on season?*

Tritium sampling occurred in all three sampling years approximately at the same season. Therefore, the observed long-term trends are unlikely to be biased by seasonal bias. In addition, the groundwaters are undoubtedly older than a few years. Hence, even if the samples were collected at different seasons, such seasonal variability in the tritium input would be smoothed out in the groundwater system integrating over time and space.

*4.1.2. I suggest being consistent with terminology 'mean transit time' or 'age', but not interchanging.*

We followed this advice and replaced "Age" with "Mean transit time" in the title of section 4.1.2.

*L425: There are other ways to generate old groundwater ages that could explain the >35 year samples. For instance, diffusion of old-aged water from rock matrix has been shown important, especially in fractured rock systems (Bethke & Johnson, 2008; Rajaram, 2021). Without the use of other tracers, confidently rejecting these old-aged samples seems like a major assumption. Some further discussion (likely in Discussion Section) seems necessary. Similarly, Eq. 4 has assumptions, namely that spring discharge is sampling flowlines across the entire storage volume, which is not always the (Berghuijs & Kirchner, 2017). I feel some discussion on this is warranted.*

See our response to Referee #1's second general comment.

*Figure 8: Can the meaning of Hdry be added to caption?*

The meaning of Hdry has been added to the caption of Figure 8.

*L569: delete 'First of all'*

We agree and have modified the manuscript accordingly.

**Response to Anonymous Referee #2**

*List of a few formal revisions. The page/line numbers refer to the annotated pdf copy of the manuscript (file hess-2023-152-ATC1).*

*Page 5*

*Line 135: at this point a very short description of the bedding types and a few words about facies types would help the reader to understand the filtration modes through the vadose and phreatic zones, also linking the regional image provided by the former statements to the very local compositional properties.*

We followed this advice. Two sentences were added here to shortly introduce the sedimentological origin of the Luxembourg Sandstone and the different subfacies related types.

*See also suggestion on lines 148-150.*

See our response to the "Lines 148-150" comment.

*Line 141: replace "chemical" with "mineralogical", or simply delete "chemical"*

We agree and have modified the manuscript accordingly.

*Lines 143-144: please clarify if these are quartzarenites with 10% calcite cement and what do You mean by "carbonate matrix". Is it true matrix, that means micrite, or do You refer to cement? What is the 60% of calcite in the calcareous sandstones? Particles? Calcareous lithoclasts? In which proportion to quartz and other lithoclasts? (watch Your classification of these rocks).*

We agree that this text is not accurate enough and have rephrased it accordingly.

*Is there some difference in porosity l.s. between the two sandstone types? I would move here the sentence from lines 153-155.*

See our response to the "Line 153-155" comment.

*Lines 148-150: this sentence could be moved at line 135, to start the description giving an idea of the stratigraphy.*

We followed this advice.

*Lines 152-153: replace permeability with hydraulic conductivity (the dimensions You use are m/sec).*

We agree and have modified the manuscript accordingly.

*Line 153-155: this sentence should be moved to the lines before the statements about hydraulic conductivity.*

We followed this advice.

*Lines 161-163: could You please relate these fracture systems to some regional structure? You mention a fault, which fault?*

We agree with this comment. We have modified the manuscript to link more clearly the fracture network and the regional geological structure.

*Page 6*

*Lines 184-185: I still disagree with this assumption, that is reinforced by the new description of the aquifer heterogeneity. As a matter of fact, 350 m/h is a very fast velocity, unusual through fractured media with low to intermediate gradients. However, this is just my opinion, based on a general consideration, hence I don't ask for changes to this statement. May be some readers would agree with me, that's all.*

*Line 224: in this list and in the new Fig.2 I count 8 districts. Which is the ninth?*

Correct, this is 8, not 9. We have modified the manuscript accordingly.